# Evaluating LLMs Across Multi-Cognitive Levels:
# From Medical Knowledge Mastery to Scenario-Based Problem Solving

**Yuxuan Zhou** [1]  **Xien Liu** [1]  **Chenwei Yan** [2]  **Chen Ning** [1]  **Xiao Zhang** [1]  **Boxun Li** [3]  **Xiangling Fu** [2]  **Shijin Wang** [4]
**Guoping Hu** [4]  **Yu Wang** [1]  **Ji Wu** [1 5 6]

## Abstract

Large language models (LLMs) have demonstrated remarkable performance on various medical benchmarks, but their capabilities across different cognitive levels remain underexplored. Inspired by Bloom's Taxonomy, we propose a multi-cognitive-level evaluation framework for assessing LLMs in the medical domain in this study. The framework integrates existing medical datasets and introduces tasks targeting three cognitive levels: preliminary knowledge grasp, comprehensive knowledge application, and scenario-based problem solving. Using this framework, we systematically evaluate state-of-the-art general and medical LLMs from six prominent families: Llama, Qwen, Gemma, Phi, GPT, and DeepSeek. Our findings reveal a significant performance decline as cognitive complexity increases across evaluated models, with model size playing a more critical role in performance at higher cognitive levels. Our study highlights the need to enhance LLMs' medical capabilities at higher cognitive levels and provides insights for developing LLMs suited to real-world medical applications.

## 1. Introduction

Large language model (LLM) technology has witnessed rapid advancement (Ouyang et al., 2022; Achiam et al., 2023; Anil et al., 2023; Touvron et al., 2023a) and shown potential in various fields, including medicine. Recently, start-of-the-art LLMs (e.g., GPT-4) have achieved expert-level performance across medical benchmarks (Singhal et al., 2023a;b; Nori et al., 2023a; Qiu et al., 2024), demonstrating their promise in real-world medical applications. However, studies (Hager et al., 2024; Yan et al., 2025) demonstrate that these models still face challenges in handling tasks that more closely resemble real-world medical scenarios, such as clinical diagnosis and treatment. This disparity raises a critical question: *how far are current LLMs from being truly applicable in real-world medical scenarios?*

Instead of directly answering this question, let's first consider a related question: *what steps are required for a medical student to become a qualified physician?* To become a qualified physician, a medical student typically undergoes a series of training stages, as shown in Figure 1 (a). First, the student acquires basic medical knowledge (e.g., anatomy, physiology, and pathology) through textbooks and lectures during the study in medical school. Then, the student learns to apply this knowledge in analyzing clinical cases during their clinical internship. Finally, the student practices diagnosing and treating patients under the guidance of experienced physicians during their residency. Such a training process is designed to align with the human cognitive process, as outlined in Bloom's Taxonomy (Anderson & Krathwohl, 2001): first memorizing and understanding knowledge, then applying it comprehensively, and finally using it to plan and solve problems in real-world scenarios.

Similar to the training process, the evaluation process in the human education system also follows the human cognitive nature. For example, a typical exam sheet includes a variety of question types (e.g., multiple-choice, short-answer, and long-response questions) that are intentionally designed to assess students' abilities across different cognitive levels. In contrast, though existing medical benchmarks provide valuable insights into LLMs' medical capabilities, they primarily evaluate LLMs' capabilities at specific cognitive levels: most medical benchmarks (Jin et al., 2021; Pal et al., 2022; Wang et al., 2024; Cai et al., 2024; Qiu et al., 2024) evaluate LLMs' capabilities through question-answering (QA) tasks, which mainly focus on assessing LLMs' preliminary knowledge grasp; some other benchmarks (Hager et al., 2024; Ouyang et al., 2024) adopt tasks such as clinical diagnosis and treatment to evaluate LLMs' capabilities of

---

[1]Department of Electronic Engineering, Tsinghua University, Beijing, China [2]School of Computer Science, Beijing University of Posts and Telecommunications, Beijing, China [3]Infinigence-AI, Beijing, China [4]iFLYTEK Research, Hefei, China [5]College of AI, Tsinghua University, Beijing, China [6]BNRist, Beijing, China. Correspondence to: Xien Liu <xeliu@mail.tsinghua.edu.cn>.

*Proceedings of the 42nd International Conference on Machine Learning*, Vancouver, Canada. PMLR 267, 2025. Copyright 2025 by the author(s).

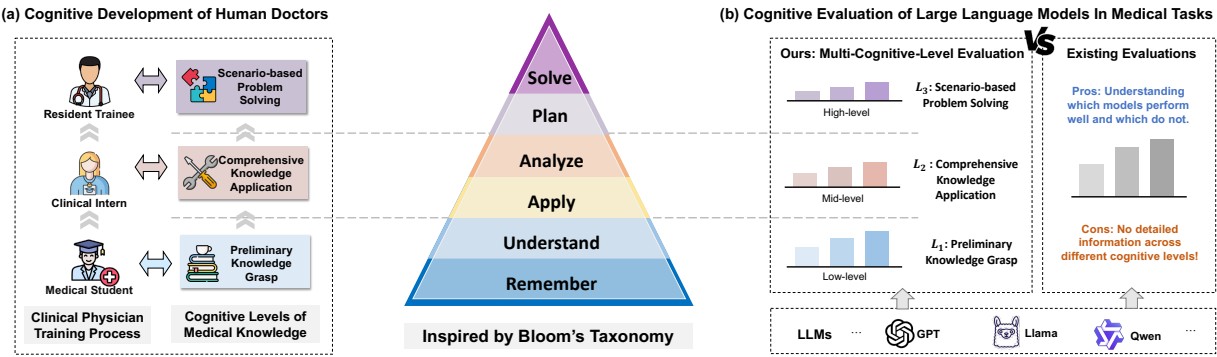

*Figure 1.* (a): The cognitive development process of human doctors; (b): Comparison of the existing evaluations and our proposed multi-cognitive-level evaluation framework (inspired by Bloom's Taxonomy) regarding cognitive levels.

scenario-based problem solving.

In this paper, we argue that the evaluation of LLMs should also follow the cognitive development process, i.e., evaluating LLMs' medical capabilities across multiple cognitive levels. To this end, we propose a **multi-cognitive-level evaluation framework** (**MultiCogEval**) to provide a comprehensive evaluation of LLMs' medical capabilities from a cognitive perspective. The schema of this framework is depicted in Figure 1 (b). Specifically, we consider three cognitive levels corresponding to the training process of human clinicians: preliminary knowledge grasp, comprehensive knowledge application, and scenario-based problem solving. Built on that, we design tasks that target each cognitive level and integrate three existing medical datasets to generate test samples for each task. To make the performance of LLMs comparable across different cognitive levels, we align the medical knowledge coverage across tasks at different levels as much as possible and normalize the performance metrics for each task.

Using this framework, we systematically evaluate existing general and medical LLMs across 2B - 70B parameters from six prominent families: Llama, Qwen, Gemma, Phi, GPT, and DeepSeek. Our findings reveal that while current SOTA LLMs generally perform well on the preliminary knowledge grasp level, their performance declines significantly as the cognitive level increases. Moreover, we find that model size plays a more crucial role in performance at higher cognitive levels. Our study provides a clear landscape of LLMs' medical capabilities across different cognitive levels and highlights the need to enhance LLMs' medical capabilities at higher cognitive levels. The codes and datasets are available at https://github.com/THUMLP/MultiCogEval. Our contributions are:

- We propose a novel evaluation framework for assessing LLMs' medical capabilities across multiple cognitive levels inspired by the human cognitive process.

- Based on the proposed framework, we systematically evaluate state-of-the-art general and medical LLMs across six prominent families.

- We reveal a significant performance decline as cognitive complexity increases across evaluated models, offering insights for developing LLMs suited to real-world medical applications.

## 2. Related Work

### 2.1. LLM Medical Evaluation Benchmarks

Recently, several medical evaluation benchmarks have been proposed to assess LLMs' medical capabilities. Most of existing medical benchmarks typically utilize the Question-answering (QA) form, where the questions are sourced from medical exams (Jin et al., 2021; Pal et al., 2022; Cai et al., 2024; Qiu et al., 2024), medical literature (Jin et al., 2019), and consumer health questions (Ben Abacha et al., 2017; Singhal et al., 2023a). Recent studies (Nori et al., 2023a;b; Singhal et al., 2023a) indicate that several LLMs perform notably on these benchmarks. For example, GPT-4 achieves an accuracy of 90.2 (with complex chain-of-thoughts prompting strategy) on the medical exam benchmark MedQA, approaching expert-level performance. Other benchmarks, such as MIMIC-IV-Ext (Hager et al., 2024) and CLIMED-Bench (Ouyang et al., 2024), adopt scenario-based tasks such as clinical diagnosis to evaluate LLMs' capabilities in solving medical problems in real-world scenarios. However, these benchmarks primarily focus on evaluating LLMs at specific cognitive levels and lack a **holistic view** of LLMs' medical capabilities across multiple cognitive levels.

### 2.2. Bloom's Taxonomy

Bloom's Taxonomy (Bloom et al., 1956) is a widely used framework for categorizing learning objectives. For learning objectives in the cognitive domain, Bloom's Taxonomy

initially proposed six levels: Knowledge, Comprehension, Application, Analysis, Synthesis, and Evaluation. In 2001, a revision of Bloom's Taxonomy (Anderson & Krathwohl, 2001) was proposed, where the six levels were renamed and reordered as follows: Remembering, Understanding, Applying, Analyzing, Evaluating, and Creating. Inspired by the Revised Bloom's Taxonomy and the training process of human clinicians, this paper introduces a multi-cognitive-level evaluation framework to assess whether LLMs have achieved the learning objectives in the medical domain across varying cognitive levels.

## 3. Methodology

### 3.1. Multi-Cognitive-Level Evaluation Priciples

We first illustrate the idea of the proposed multi-cognitive-level evaluation. Given a large language model $\mathcal{M}$ and a series of cognitive levels $L_1, L_2, \ldots, L_n$, the goal of the multi-cognitive-level evaluation is to assess the capabilities of $\mathcal{M}$ at each cognitive level:

$$\mathbf{f}(\mathcal{M}) = [f_1(\mathcal{M}), f_2(\mathcal{M}), \ldots, f_n(\mathcal{M})] \qquad (1)$$

where $\mathbf{f}$ refers to the whole multi-cognitive-level evaluation process, $f_i$ represents the evaluation process at the $i$-th cognitive level, and $f_i(\mathcal{M})$ denotes the corresponding evaluation results.

A qualified multi-cognitive-level evaluation should also adhere to the following principles:

- **Task Relevance**: The evaluation tasks should target different cognitive levels, aligning with the knowledge-learning process.

- **Knowledge Consistency**: In order to isolate the effects of cognitive levels, it is important to ensure that the evaluation tasks have consistent knowledge coverage across different cognitive levels, minimizing the influence of knowledge coverage on the evaluation results.

- **Metric Alignment**: The performance metrics should be aligned (normalized) across different cognitive levels to ensure the comparability of the evaluation results.

In our study, we adopt these principles to develop a multi-cognitive-level evaluation framework for assessing LLMs' medical capabilities. Though we primarily focus on the medical domain, the proposed multi-cognitive-level evaluation principles can also be generalized to other domains.

### 3.2. Cognitive Levels in Medical Domain

Following the multi-cognitive-level evaluation principles, we first define three cognitive levels for assessing LLMs'

medical capabilities by aligning with the training process of human clinicians and referring to Bloom's Taxonomy. The three cognitive levels are as follows[1]:

- **Preliminary Knowledge Grasp** ($L_1$): The first level corresponds to the education of medical students in medical school, where they learn and grasp basic medical knowledge through textbooks and lectures. At this level, we focus on evaluating LLMs' capabilities in memorizing and initially understanding medical knowledge that is essential for medical practice.

- **Comprehensive Knowledge Application** ($L_2$): The second level relates to the clinical internship stage, where students learn to apply basic medical knowledge to handle simple clinical cases. This level aims to evaluate LLMs' capabilities in applying medical knowledge to analyze and solve complex medical problems.

- **Scenario-based Problem Solving** ($L_3$): This level corresponds to the residency training stage, where students practice advanced clinical skills (e.g., diagnosis, treatment) under the guidance of experienced physicians. At this level, we aim to evaluate LLMs' capabilities in planning and solving problems in real-world medical scenarios using medical knowledge.

### 3.3. Multi-Cognitive-Level Evaluation Framework

Based on the defined cognitive levels, we develop a multi-cognitive-level evaluation framework that integrates existing medical datasets and designs tasks tailored to each cognitive level. The overview of the proposed framework is illustrated in Figure 2. In the following sections, we will introduce the detailed design of the evaluation tasks, dataset construction, and performance metrics that follow the multi-cognitive-level evaluation principles proposed in Section 3.1.

**Task Design**  For Low-Level tasks ($f_1$), we adopt multiple-choice questions (**MCQs**) as simple knowledge recalling tasks for Low-Level, where the model is required to recognize the correct answer from multiple candidate options (see the left part of Figure 2). Such task format is commonly used in existing medical benchmarks and human examinations to evaluate students' basic knowledge understanding.

For Mid-Level tasks ($f_2$), given that existing medical benchmarks fail to adequately address this cognitive level—either being too simplistic, as in QA-based benchmarks, or overly complex, as in scenario-based benchmarks—we reformulate the original MCQs into a set of "**Complex Knowledge Application**" tasks for this level. These tasks are designed to assess the cognitive skills required for real-world medical

---

[1]In the following sections, we refer to these levels as *Low-Level, Mid-Level, and High-Level* for simplicity.

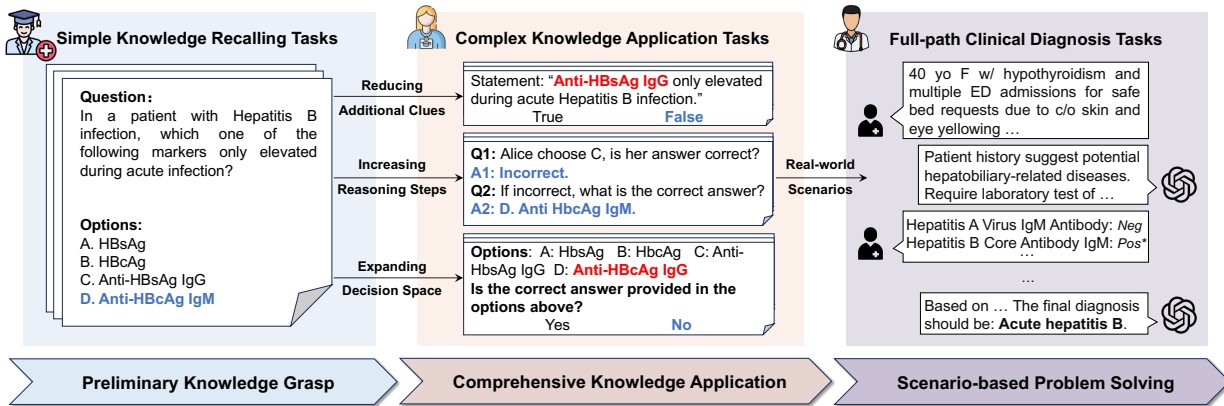

*Figure 2.* An overview of the proposed multi-cognitive-level medical evaluation framework.

tasks from different perspectives without relying on specific scenarios, thereby effectively evaluating the model's ability in the Mid-Level. Specifically, we consider the following task types (see the middle part of Figure 2):

- **Statement Validation Questions**: We transform the question and one of the candidate options from the original MCQs into a statement, requiring the model to determine whether the statement is true or false. While this task may intuitively seem simpler than MCQs, the answer choices in MCQs can provide additional clues, allowing the model to choose the most plausible answer without fully understanding why it is correct. In contrast, statement validation emphasizes the model's ability to precisely discriminate knowledge, closely mirroring real-world clinical scenarios where options are typically not provided. Furthermore, given that some MCQ distractors significantly increase task difficulty, we construct two statements per MCQ: one reflecting the correct answer and the other derived from a distractor, maintaining their difficulty.

- **Multi-Step Rectification Questions**: Handling QA tasks typically requires a single decision-making step, whereas problem-solving in real-world scenarios often involves multiple steps. For instance, when managing a patient's care, a doctor first establishes a diagnosis and then prescribes treatment based on the diagnosis made in the previous step. Inspired by this, we reformulate the original MCQ into a two-step task: we first ask the model to verify whether a provided option is the correct answer of the MCQ, and then, if the answer is incorrect, we ask the model to give the correct answer from the rest of options. We generate two questions for each MCQ: one provides the correct answer, while the other provides a randomly sampled wrong option.

- **Answer Existence Judgment Questions**: QA tasks typically operate within a constrained decision space

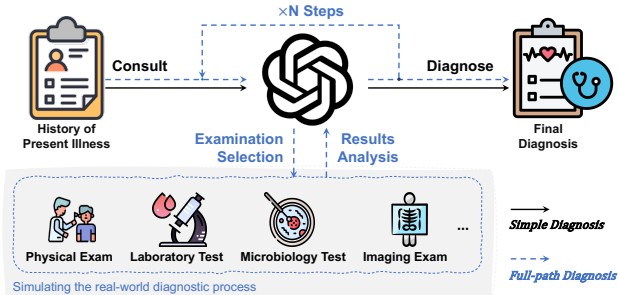

*Figure 3.* An overview of the full-path clinical diagnosis task applied in the proposed evaluation framework.

(e.g., predefined answer choices), whereas real-world tasks involve significantly broader decision spaces. For instance, there may be hundreds of potential candidate diseases in clinical diagnosis, requiring more complex decision-making to reach the correct diagnosis. To simulate this, we design a task where the model is asked to determine whether the correct answer exists in a set of candidate options. For each original MCQ, we generate a pair of questions with opposite labels to eliminate the influence of random guessing: one question retains the original options, while the correct option in the other question is replaced with a distractor.

For High-Level tasks ($f_3$), we follow Hager et al. (2024) and assess LLMs' capabilities of solving scenario-based problems using a full-path clinical diagnosis task, as illustrated in Figure 3. Compared to the tasks in the Low-Level and Mid-Level, this task **simulates a real-world diagnostic process** by requiring the model to sequentially order examinations and integrate information from the obtained examination results to arrive at a final diagnosis. Solving this task requires models to perform multi-step decision-making within a larger decision space, mirroring the complexity of

**Recognize & Retrieve Procedure for High-level Dataset Construction**

*Figure 4.* The recognize-and-retrieve procedure for constructing the High-Level dataset, ensuring alignment with previous levels in terms of knowledge coverage.

real-world medical scenarios.

**Dataset Construction** For Low-Level tasks, we directly use the questions from the MedQA (Jin et al., 2021) and MedMCQA (Pal et al., 2022) datasets, which cover a wide range of medical knowledge and are widely used to evaluate LLMs' medical capabilities. For Mid-Level tasks, as introduced above, we reformulate each original MCQ into two statement validation questions, two multi-step rectification questions, and two answer existence judgment questions. We apply the advanced GPT-4o to assist in data generation. Specifically, GPT-4o is employed to (1) generate statements based on the questions and the provided options for the statement validation tasks and (2) generate the distractors for the answer existence judgment tasks based on the original question and the correct answer. We arrange for three doctors with over three years of experience to verify the statements and distractors generated by GPT-4o and find that the error rate of GPT-4o is less than 5%.

For High-Level tasks, we leverage the MIMIC-IV dataset (Johnson et al., 2023), which contains detailed electronic health records (EHRs) of patients, including the history of present illness, physical examination, laboratory tests, microbiology tests, and imaging tests. We employ a *recognize-and-retrieve* procedure to construct the dataset, ensuring its knowledge coverage remains consistent with previous levels (see Figure 4). Specifically, we first use the medical NER tool MedCAT (Kraljevic et al., 2021) to identify diseases in previous-level tasks and construct a disease pool. We then filter the admission records associated with these diseases using ICD-10-CM codes, which are widely employed for disease classification. To ensure alignment between primary diagnoses and the disease pool, we extract admission records where the primary discharge diagnosis matches a disease in the pool. Records with more than two diseases in the discharge diagnosis are excluded to maintain a one-to-one mapping between records and diseases. After processing, 2,176 admission records remain, covering 42 diseases across eight human body systems. Finally, we convert these records into test samples for the full-path clinical

*Table 1.* The basic statistics of the constructed multi-cognitive-level medical evaluation benchmark.

| Levels | Tasks | Sources | # Samples |
|---|---|---|---|
| Low | MCQs | MedQA | 795 |
| | | MedMCQA | 2,816 |
| Middle | Reformulated Tasks | MedQA | 4,770 |
| | | MedMCQA | 16,896 |
| High | Scenario-based Diagnosis | MIMIC-IV | 2,176 |

diagnosis task. Table 1 summarizes the key statistics of the constructed multi-cognitive-level evaluation benchmark, while additional details, including task examples, prompts, and answer parsing, are provided in Appendix A.

**Performance Metrics** We adopt the accuracy for the Low-Level and Mid-Level tasks. For the High-Level task, we propose *full-path diagnosis accuracy*, a new metric that assesses LLMs' diagnostic performance by considering both procedural correctness and outcome accuracy:

$$\text{Accuracy}_{\text{proc}} = \frac{1}{N} \sum_{i=1}^{N} \mathbb{I}(\text{Diag}_{\text{pred}}^{(i)} = \text{Diag}_{\text{gt}}^{(i)}) \times \text{Recall}_{\text{exam}}^{(i)}$$

(2)

where $\text{Diag}_{\text{pred}}^{(i)}$ and $\text{Diag}_{\text{gt}}^{(i)}$ denote the predicted and ground-truth diagnoses of the $i$-th admission record, respectively. $\mathbb{I}(\cdot)$ is the indicator function, and $N$ is the number of admission records. $\text{Recall}_{\text{exam}}^{(i)}$ is the examination recall of the $i$-th admission record, which is calculated as:

$$\text{Recall}_{\text{exam}}^{(i)} = \frac{\text{\# Correct Ordered Exam Items}}{\text{\# Total Exam Items in the Record}} \quad (3)$$

In this study, we prioritize examination recall, as missing critical examination information can have far more severe consequences than ordering non-essential tests during diagnosis. To ensure fairness, we compute the **macro-average** of both examination recall (across examination types) and diagnosis accuracy (across diseases), assuming equal importance for each examination type and disease in this task.

Considering that the random guessing performance may vary across different tasks, we also calculate *normalized accuracy* of each task by reducing the effect of random guessing to ensure the comparability of the evaluation results across different cognitive levels:

$$\text{Accuracy}_{\text{norm}} = \frac{\text{Accuracy}_{\text{model}} - \text{Accuracy}_{\text{rand}}}{1 - \text{Accuracy}_{\text{rand}}} \quad (4)$$

More details about the metrics and normalization process are provided in Appendix B.

*Table 2.* Performance (mean and standard error of the normalized accuracy (%)) of LLMs across different cognitive levels evaluated on the proposed benchmark. Llama1's performance on the High-Level task is unavailable due to the absence of instruction-tuned versions. Asterisks: due to the high cost of GPT-4o and DeepSeek-V3 API, we evaluated it on ∼10% of the original dataset.

| Model | Low-Level | | Mid-Level | | High-Level |
| | MedQA | MedMCQA | MedQA | MedMCQA | MIMIC-IV |
| --- | --- | --- | --- | --- | --- |
| Llama-7B | 3.36 ± 0.82 | 10.40 ± 0.24 | 2.47 ± 0.76 | 4.38 ± 0.62 | - |
| Llama-13B | 13.18 ± 0.57 | 18.98 ± 0.22 | 2.62 ± 0.65 | 3.15 ± 0.71 | - |
| Llama-33B | 24.09 ± 0.26 | 26.88 ± 0.34 | 12.85 ± 0.62 | 10.44 ± 0.19 | - |
| Llama-65B | 26.64 ± 0.49 | 29.59 ± 0.29 | 16.82 ± 0.73 | 13.48 ± 0.44 | - |
| Llama2-7B | 15.88 ± 0.28 | 18.32 ± 0.22 | 4.21 ± 1.01 | 5.40 ± 0.55 | 2.65 ± 0.20 |
| Llama2-13B | 21.10 ± 0.30 | 20.94 ± 0.33 | 6.18 ± 1.25 | 7.44 ± 0.29 | 4.92 ± 0.15 |
| Llama2-70B | 37.99 ± 0.34 | 35.60 ± 0.26 | 20.92 ± 0.80 | 16.12 ± 0.53 | 5.16 ± 0.18 |
| Llama3-8B | 37.30 ± 0.53 | 39.11 ± 0.29 | 11.47 ± 0.43 | 11.40 ± 0.28 | 10.50 ± 0.16 |
| Llama3-70B | 63.68 ± 0.34 | 60.82 ± 0.28 | 41.79 ± 0.85 | 36.25 ± 0.46 | 17.81 ± 0.41 |
| Qwen-7B | 19.40 ± 0.31 | 24.85 ± 0.21 | 7.34 ± 0.39 | 7.08 ± 0.32 | 2.51 ± 0.31 |
| Qwen-14B | 38.05 ± 0.38 | 39.25 ± 0.13 | 17.98 ± 0.63 | 15.98 ± 0.32 | 4.14 ± 0.29 |
| Qwen-72B | 50.60 ± 0.39 | 50.95 ± 0.18 | 28.39 ± 0.70 | 26.28 ± 0.31 | 5.89 ± 0.35 |
| Qwen2-7B | 36.73 ± 0.44 | 39.85 ± 0.15 | 21.22 ± 0.51 | 20.93 ± 0.18 | 6.99 ± 0.25 |
| Qwen2-72B | 65.91 ± 0.24 | 60.41 ± 0.19 | 47.32 ± 0.43 | 37.19 ± 0.18 | 15.10 ± 0.09 |
| Qwen2.5-7B | 42.86 ± 0.49 | 45.39 ± 0.17 | 27.01 ± 0.21 | 24.75 ± 0.17 | 9.50 ± 0.04 |
| Qwen2.5-14B | 52.23 ± 0.30 | 51.74 ± 0.24 | 35.62 ± 0.37 | 30.73 ± 0.31 | 12.07 ± 0.21 |
| Qwen2.5-32B | 59.21 ± 0.21 | 57.17 ± 0.09 | 38.45 ± 0.56 | 33.43 ± 0.24 | 11.87 ± 0.22 |
| Qwen2.5-72B | 67.83 ± 0.20 | 61.82 ± 0.12 | 49.47 ± 0.69 | 40.57 ± 0.39 | 16.05 ± 0.24 |
| Gemma-2B | 6.16 ± 0.63 | 15.04 ± 0.24 | 0.52 ± 0.31 | 2.82 ± 0.31 | 0.59 ± 0.09 |
| Gemma-7B | 30.35 ± 0.42 | 27.41 ± 0.38 | 15.09 ± 0.48 | 13.64 ± 0.37 | 0.33 ± 0.09 |
| Gemma2-2B | 14.87 ± 0.34 | 24.18 ± 0.31 | 1.78 ± 0.71 | 3.85 ± 0.55 | 5.02 ± 0.33 |
| Gemma2-9B | 44.21 ± 0.18 | 46.20 ± 0.20 | 24.52 ± 0.28 | 23.05 ± 0.24 | 9.04 ± 0.28 |
| Gemma2-27B | 53.65 ± 0.31 | 50.45 ± 0.18 | 33.40 ± 0.88 | 30.33 ± 0.26 | 11.19 ± 0.64 |
| Phi3-3.8B | 37.52 ± 0.28 | 40.96 ± 0.20 | 23.10 ± 1.15 | 22.53 ± 0.37 | 2.21 ± 0.28 |
| Phi3-7B | 49.25 ± 0.30 | 46.95 ± 0.30 | 28.87 ± 0.31 | 21.75 ± 0.28 | 7.74 ± 0.16 |
| Phi3-14B | 54.06 ± 0.49 | 49.70 ± 0.27 | 31.67 ± 0.55 | 28.46 ± 0.24 | 3.89 ± 0.01 |
| Phi4-14B | 56.19 ± 0.40 | 52.62 ± 0.30 | 39.30 ± 0.42 | 33.46 ± 0.23 | 12.18 ± 0.37 |
| GPT-4o-mini | 62.67 ± 0.20 | 55.33 ± 0.06 | 42.98 ± 0.59 | 32.28 ± 0.66 | 13.81 ± 0.84 |
| GPT-4o* | 78.33 ± 0.75 | 64.00 ± 1.02 | **56.96 ± 1.84** | **41.65 ± 1.59** | 19.33 ± 0.30 |
| DeepSeek-V3* | **78.33 ± 0.75** | **65.78 ± 0.22** | 44.79 ± 0.30 | 40.07 ± 1.49 | **19.42 ± 0.17** |

# 4. Experiments

## 4.1. Experimental Setup

**Evaluated Models** To conduct a systematic evaluation on current LLMs, we select a total of 32 general LLMs across six LLM families, including Llama (Touvron et al., 2023a;b; Dubey et al., 2024), Qwen (Bai et al., 2023; Yang et al., 2024; Hui et al., 2024), Gemma (Mesnard et al., 2024; Team et al., 2024), Phi (Abdin et al., 2024b;a), GPT-4o (OpenAI, 2024), and DeepSeek (Liu et al., 2024). We also evaluate 8 medical LLMs from 4 series: ClinicalCamel (Toma et al., 2023), Med42 (v1, v2) (Christophe et al., 2024a;b), Meditron (Chen et al., 2023), and MMed-Llama (Qiu et al., 2024). The detailed statistics are provided in Appendix C.

**Evaluation Setting** For tasks in the Low-Level and Mid-Level, we leverage the five-shot in-context learning (Brown et al., 2020) for evaluation, where five input-output pairs are sampled as the demonstrative examples to guide the model to generate the correct answer for the test sample (see Figure 9 in Appendix A). For the High-Level task (full-path

diagnosis), as it involves multi-turn interaction, we evaluate the instruction fine-tuned versions of the corresponding models and leverage the zero-shot learning setting to evaluate the models' performance. Specifically, for each test sample, we provide the model with the instructions for the task and ask it to order the specific exam items sequentially and make the final diagnosis. We utilize the UMLS Metathesaurus (Bodenreider, 2004) along with a handcrafted synonym set to identify the model's predictions for ordered exam items and the final diagnosis (the detection rate across all categories reached ∼95%). We conduct repeated evaluations 3-5 times and report the average performance and standard error. More details about the evaluation setting (e.g., hyperparameters) are provided in Appendix D.

## 4.2. Cognitive-Level Analysis

**Higher the Cognitive Level, Worse the Model Performance** We first compare the performance of LLMs across different cognitive levels on the proposed benchmark and present the results in Table 2. For the Mid-Level, we use the

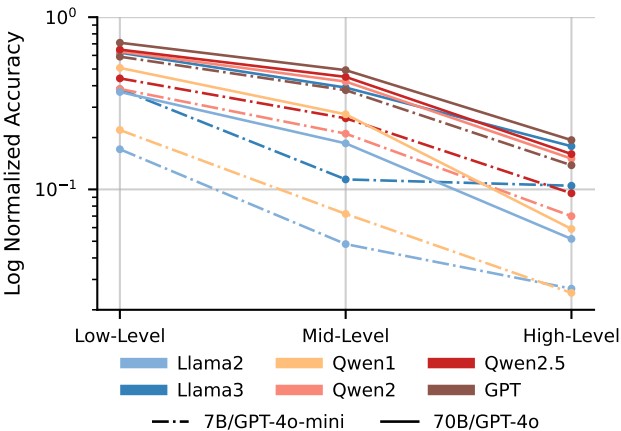

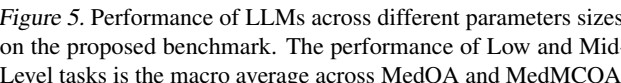

*Figure 5.* Performance of LLMs across different parameters sizes on the proposed benchmark. The performance of Low and Mid-Level tasks is the macro average across MedQA and MedMCQA.

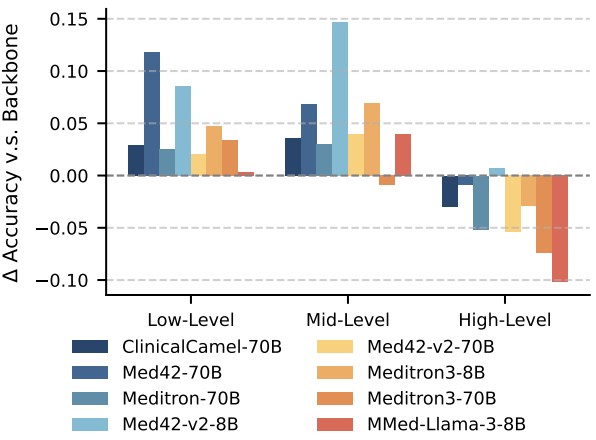

*Figure 6.* Performance gains of medical LLMs over their backbone models across different cognitive levels.

average performance of the three task types to compare with the other two levels (Detailed performance is provided in Table 8 of Appendix E). We observe that several state-of-the-art LLMs, such as GPT-4o, DeepSeek-V3, and Llama3-70B, achieve remarkable performance (>60%) on the Low-Level (preliminary knowledge grasp) tasks. However, all of these models experience a significant performance drop (~20%) when evaluated on the Mid-Level tasks (comprehensive knowledge application). Moreover, on High-Level tasks (scenario-based problem solving), the performance of all models further declines, with the best-performing model, DeepSeek-V3, achieving only 19.4 full-path diagnosis accuracy. This indicates that while current LLMs grasp basic medical knowledge well, they still face significant challenges at higher cognitive levels, particularly when solving complex problems in real-world medical scenarios.

**Parameter Sizes Matter, Especially at Higher Cognitive Levels**   We further investigate the impact of model parameter sizes across different cognitive levels. Specifically, we examine 7B and 70B models from the Llama and Qwen families, along with GPT-4o and GPT-4o-mini[2]. As shown in Figure 5, larger models consistently outperform their smaller counterparts within the same model series. Additionally, the performance gap between model sizes increases significantly from the Low-Level to the Mid-Level on a logarithmic scale, suggesting that model size plays a more critical role in enabling LLMs to tackle complex problem-solving tasks. However, this gap narrows slightly at the High-Level, implying that even the 70B-level models struggle to effectively solve scenario-based problems.

[2]DeepSeek-V3 is not involved in this analysis since it does not have a smaller version for comparison.

*Table 3.* Performance (%) of LLMs fine-tuned with/without inference-time scaling. The performance of Low and Mid-Level tasks is the macro average across MedQA and MedMCQA.

| Model | Low-Level | Mid-Level | High-Level |
|---|---|---|---|
| DeepSeek-V3 | 72.1 | 42.4 | 19.4 |
| DeepSeek-R1 | **81.9** | **65.5** | **26.5** |
| GPT-4o-mini | 59.0 | 37.7 | 13.8 |
| o3-mini | **81.3** | **64.5** | **15.1** |

**Medical-Domain Finetuning Primarily Benefits Low- and Mid-Level Tasks**   To investigate the impact of medical-domain finetuning on LLMs across different cognitive levels, we compare the performance of medical LLMs with their backbone models. As shown in Figure 6, these specialized models generally outperform their backbone counterparts on Low- and Mid-Level tasks, with performance gains reaching up to 15%. However, they fail to achieve significant improvements on High-Level tasks and even underperform their backbone models. This phenomenon indicates that while medical-domain finetuning strengthens LLMs' grasp of basic medical knowledge and its comprehensive application, it remains less effective in addressing complex real-world medical scenarios.

**Inference-time Scaling Works Well, Especially for Higher Levels**   We further investigate the effectiveness of inference-time scaling on LLMs' medical capabilities across different cognitive levels. We select two representative models with varied parameter scales (DeepSeek-V3 and GPT-4o-mini) and their corresponding inference-time scaling versions (DeepSeek-R1 and o3-mini) for study. As presented in Table 3, the inference-time scaling models consistently outperform their backbone models across all cog-

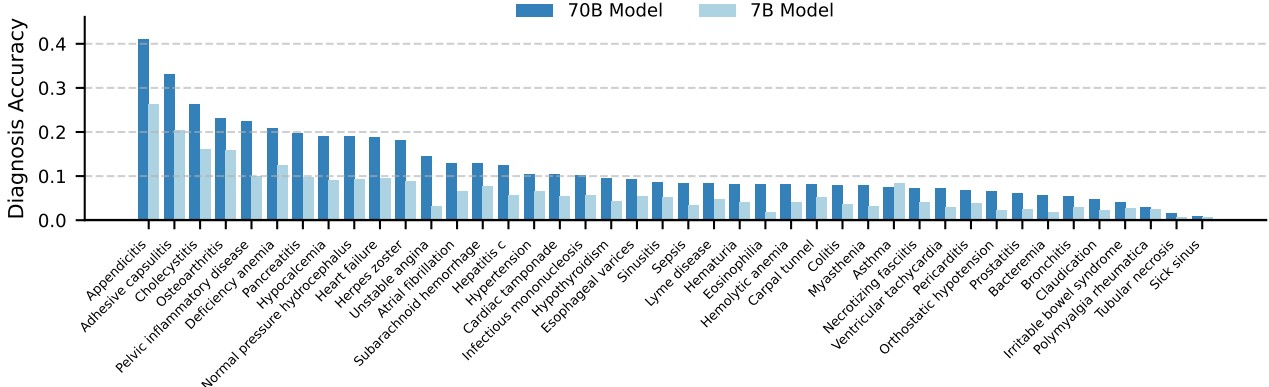

*Figure 7.* Average diagnosis accuracy of LLMs (Llama2 and 3, Qwen1, 2, and 2.5) across diseases in the full-path clinical diagnosis task.

*Table 4.* Detailed performance (%) of LLMs on tasks in the Mid-Level. StmtVal: Statement Validation; AnsExist: Answer Existence Judgment; MultiRect: Multi-Step Rectification.

| Model | Low | Mid-Level | | |
|---|---|---|---|---|
| | | StmtVal | AnsExist | MultiRect |
| Llama3-70B | 62.3 | 47.2 | 9.9 | 60.0 |
| Qwen2.5-72B | 64.8 | 50.2 | 23.8 | 61.0 |
| Gemma2-27B | 52.1 | 38.5 | 10.4 | 46.7 |
| Phi4-14B | 54.4 | 40.1 | 17.5 | 51.6 |
| GPT-4o | 71.2 | 58.8 | 22.4 | 66.3 |
| DeepSeek-V3 | 72.1 | 53.7 | 6.3 | 67.4 |

*Table 5.* Detailed performance (%) of LLMs on the full-path clinical diagnosis task. Exam Recall: the macro average of examination recall across examination types. End-point: the accuracy of the final diagnosis without considering the examination recall.

| Model | Exam Recall | Diagnosis Accuracy | |
|---|---|---|---|
| | | End-point | Full-path |
| Llama3-70B | 38.8 | 38.1 | 18.1 |
| Qwen2.5-72B | 34.4 | 39.6 | 15.8 |
| Gemma2-27B | 27.5 | 33.4 | 10.6 |
| Phi4-14B | 27.1 | 39.2 | 12.2 |
| GPT-4o | 31.8 | 49.2 | 19.3 |
| DeepSeek-V3 | 30.0 | 53.6 | 19.4 |

nitive levels, while they achieve more significant improvement on Mid-Level tasks compared to Low-Level tasks (e.g., +23.1 vs. +9.8 for DeepSeek-R1). Note that the performance gap narrows on high-level tasks due to their significantly increased difficulty. This suggests that inference-time scaling is a promising approach to enhance LLMs' medical capabilities, particularly for complex problem-solving tasks.

### 4.3. Fine-Grained Analysis

**LLMs Perform Constantly Worse Across Tasks in the Mid-Level** We further provide a fine-grained analysis of LLMs' performance across tasks within the Mid-Level and compare it with the Low-Level tasks. We illustrate results of the best-performed model from each LLM family in Table 4, with the full results presented in Table 8. We observe that LLMs consistently perform worse on all Mid-Level tasks compared to Low-Level tasks. Specifically, the performance drop on the answer existence judgment tasks is more significant, where almost all models experience a performance decrease of over 40%. This suggests that current LLMs struggle with medical problems involving significantly larger decision spaces, which is a key requirement in

real-world medical scenarios.

**Low Procedural Correctness Causes Bad Performance in Scenario-based Tasks** We further analyze the performance of large language models (LLMs) on the full-path clinical diagnosis task. As shown in Table 5, LLMs exhibit moderate performance in terms of end-point accuracy (i.e., diagnosis accuracy without considering examination recall), with DeepSeek-V3 achieving the highest accuracy at 53.6%. However, when taking examination recall into consideration (full-path performance), the accuracy of all models declines significantly, with the best-performing model, DeepSeek-V3, achieving only 19.4% diagnosis accuracy. This decline can primarily be attributed to low examination recall (less than 40%) across all models, suggesting that while current LLMs can occasionally reach the correct diagnosis, they fail to fully consider all relevant possibilities and request the necessary information throughout the diagnostic process.

**LLM Performance Across Diseases Presents Long-Tail Distribution** Finally, we analyze the diagnosis performance of LLMs across diseases in the full-path clinical

*Table 6.* Clinician validation results on the proposed benchmark. Clinician Acc.: the percentage of correct answers provided by the clinicians. Clinician Subj. Diff.: the subjective difficulty by clinicians on a scale of 1-10 (1: very easy, 10: very difficult).

| Level | Clinician Acc. (%) | Clinician Subj. Diff. | |
|---|---|---|---|
| | | Mean | Median |
| Low | 68.8 | 5.0 | 5.5 |
| Mid | 54.2 | 6.0 | 5.8 |
| High | 23.5 | 7.5 | 7.5 |

diagnosis task and present the results in Figure 7. We observe that the performance of LLMs varies significantly across different diseases, forming a long-tail distribution, while 70B models generally perform better than 7B models across all diseases. This suggests that current LLMs may struggle with diagnosing rare or complex diseases, which are critical in real-world medical scenarios.

### 4.4. Clinician Validation

Finally, to ensure that the tasks at different levels of the proposed benchmark genuinely correspond to distinct levels of cognitive difficulty, we further conduct a small-scale clinician validation study. Specifically, we randomly select 100 samples from the constructed benchmark and recruit four licensed clincians with 3 to 8 years of experience to assess the benchmark's difficulty from two perspectives: (1) their accuracy in answering the questions and (2) their subjective difficulty ratings to the questions on a scale from 1 (Easy) to 10 (Hard). The evaluation results are provided in Table 6. We found that clinicians' accuracy also decreases as the cognitive level increases across three levels, indicating that tasks at higher levels are indeed more challenging. Moreover, their subjective difficulty ratings align with this trend, further demonstrating the validity of the proposed benchmark. More details about the clinician validation study, including the data selection and evaluation process, are provided in Appendix F.

### 5. Conclusion

In this work, we propose a multi-cognitive-level evaluation framework that assesses LLMs' medical capabilities at three cognitive levels. Using the proposed framework, we construct a new medical benchmark and systematically evaluate existing LLMs on the benchmark. Our study leads to the following key findings: (1) While the performance of smaller LLMs (∼10B) gradually approaches that of larger LLMs (>70B) on Low-Level tasks, the performance gap remains significant on Mid-Level and High-Level tasks, indicating that model size plays a crucial role in enabling LLMs to tackle complex medical problems; (2) Medical-domain finetuning significantly improves performance on

Low- and Mid-Level tasks but has limited impact on High-Level tasks. This suggests that current medical-specific finetuning strategies are insufficient for enhancing LLMs' reasoning abilities in complex real-world medical scenarios; (3) Inference-time scaling shows promise in boosting LLMs' medical capabilities, especially in tasks requiring complex knowledge application. However, further research is required to enhance LLMs' High-Level capabilities, such as planning, requesting key information, and reasoning based on the acquired information.

Based on these findings, we offer the following insights for applying and developing large language models to address real-world clinical challenges: (1) **Model Selection**: For low-level medical tasks such as knowledge-based question answering, LLMs with around 10B parameters are generally sufficient. However, for more complex clinical tasks—such as diagnosis and treatment recommendation—larger LLMs are necessary; (2) **Medical LLM Development**: Future medical-domain finetuning efforts should focus more on enhancing LLMs' High-Level cognitive abilities, including clinical planning, proactively acquiring key diagnostic and treatment information, and conducting multi-step reasoning. Inference-time scaling emerges as a promising direction to support these high-level capabilities.

To the best of our knowledge, this work is the first to evaluate LLMs' medical capabilities across multiple cognitive levels. While promising, the evaluation may be constrained by the scope of medical knowledge coverage and task diversity. Future research could further explore these areas, expanding the range of medical domains and tasks to offer a more holistic view of LLMs' medical abilities.

### Acknowledgements

This work was supported by the Noncommunicable Chronic Diseases-National Science and Technology Major Project (Grant No. 2023ZD0506501) and Beijing Natural Science Foundation (NO. 4252046). We would like to thank the clinicians who participated in the validation and the anonymous reviewers for their valuable feedback.

### Impact Statement

This work mainly explores a multi-cognitive-level evaluation framework for assessing LLMs' medical capabilities. The results indicate that current LLMs generally perform well on the preliminary knowledge grasp but face challenges at higher cognitive levels, providing insights for developing LLMs suited to real-world medical applications. The datasets and codes used in this study are built based on public medical datasets and are publicly available to facilitate further research. We have not identified any potential negative ethical consequences requiring further consideration.

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

# A. Details of Datasets

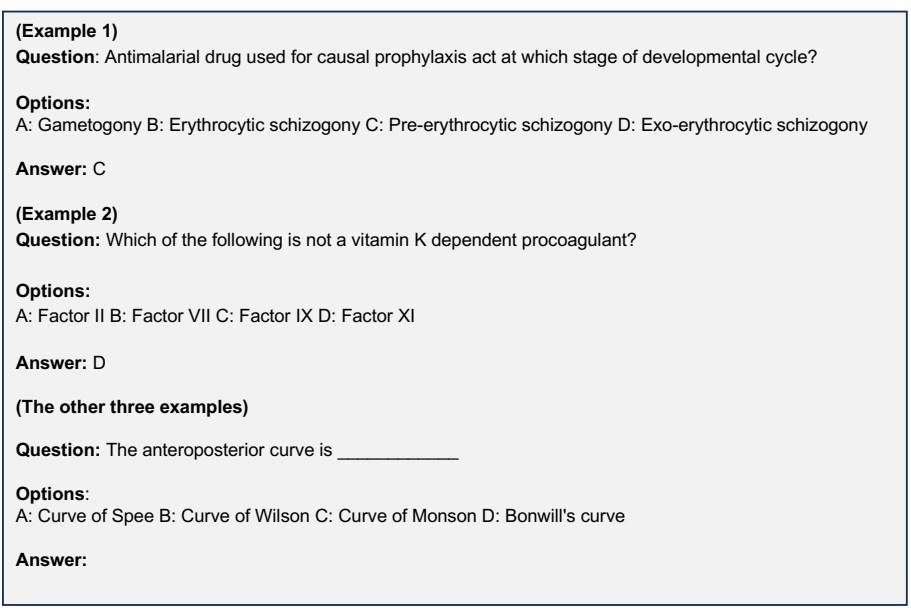

**Question:** A 25-year-old man comes to the office because of pain in his left shoulder. He says that this pain started 3 years ago … Which of the following enzymes is most likely deficient in this patient?

**Options:**
**A:** Branched-chain alpha-ketoacid dehydrogenase
**B:** Cystathionine synthase deficiency
**C:** Homogentisic acid oxidase
**D:** Phenylalanine hydroxylase
**E:** Propionyl-CoA carboxylase

**Answer:** C

**Question:** Antimalarial drug used for causal prophylaxis act at which stage of developmental cycle?

**Options:**
A: Gametogony
B: Erythrocytic schizogony
C: Pre-erythrocytic schizogony
D: Exo-erythrocytic schizogony

**Answer:** C

*Figure 8.* Examples of Low-Level tasks. Left: a MedQA question. Right: a MedMCQA question.

## A.1. Low-Level Tasks

As introduced in Section 3.3, we adopt the original multiple-choice questions in the MedQA and MedMCQA datasets for Low-Level tasks. MedQA is a large-scale medical exam dataset, containing medical exam questions sourced from three different regions. In this study, we use the questions in the US subset, which contains five-option multiple-choice questions collected from the United States Medical Licensing Examination (USMLE). MedMCQA is another large-scale medical exam dataset that contains multiple-choice questions sourced from All India Institute of Medical Sciences Entrance Examination (AIIMS) and National Eligibility cum Entrance Test for Postgraduate (NEET-PG) in India. The questions in MedMCQA are four-option MCQs. The language used in both datasets is English. For MedQA, we use the test set for evaluation; for MedMCQA, we use the dev set, because the ground truth of MedMCQA's test set is not available to the public. We filtered out the MedMCQA questions that are marked as multiple correct answers. As a result, we have 800 questions from MedQA and 2,816 questions from MedMCQA for Low-Level tasks. Examples of Low-Level tasks are shown in Figure 8.

---

**(Example 1)**
**Question**: Antimalarial drug used for causal prophylaxis act at which stage of developmental cycle?

**Options:**
A: Gametogony B: Erythrocytic schizogony C: Pre-erythrocytic schizogony D: Exo-erythrocytic schizogony

**Answer:** C

**(Example 2)**
**Question:** Which of the following is not a vitamin K dependent procoagulant?

**Options:**
A: Factor II B: Factor VII C: Factor IX D: Factor XI

**Answer:** D

**(The other three examples)**

**Question:** The anteroposterior curve is ____________

**Options**:
A: Curve of Spee B: Curve of Wilson C: Curve of Monson D: Bonwill's curve

**Answer:**

---

*Figure 9.* Input prompt format of the five-shot in-context learning setting.

We use five-shot in-context learning to evaluate the performance of LLMs on Low-Level tasks. Figure 9 shows the input prompt format of the five-shot in-context learning setting. The input prompt consists of five examples, each of which is a question-answer pair. After the five examples, we append the test sample, which is a question without the correct answer. The model is required to predict the correct answer for the test sample. For answer parsing, we find that current LLMs

always follow the correct format in the five-shot in-context learning setting. Therefore, we directly compare the generated answer with the ground truth answer.

## A.2. Mid-Level Tasks

| | | |
|---|---|---|
| **Question:** A 25-year-old man comes to the office because of pain in his left shoulder. because of pain in his left shoulder. He says that this pain started 3 years ago and has progressively worsened. He denies joint trauma, fever, dysuria, or morning stiffness. … Statement: "Homogentisic acid oxidase is most likely deficient in this patient", is the statement above correct or incorrect? | **Question:** A 25-year-old man comes to the office because of … Which of the following enzymes is most likely deficient in this patient?
**Options:**
**A:** Branched-chain alpha-ketoacid dehydrogenase
**B:** Cystathionine synthase deficiency
**C:** Homogentisic acid oxidase
**D:** Phenylalanine hydroxylase
**E:** Propionyl-CoA carboxylase
Alice chose answer D. Please verify if this is correct, and if not, provide the correct answer. | **Question:** A 25-year-old man …There is a 50% chance that the correct answer is not included in the options. Please determine if the correct answer is among the given options and respond with yes or no.

**Options:**
A: Branched-chain alpha-ketoacid dehydrogenase
B: Cystathionine synthase
C: Tyrosine aminotransferase (distractor)
D: Phenylalanine hydroxylase
E: Propionyl-CoA carboxylase |
| **Answer:** correct | **Answer:** incorrect, answer C is the correct answer | **Answer:** no |

*Figure 10.* Examples of Mid-Level tasks. Left: Statement Validation tasks; Middle: Multi-step Rectification tasks; Right: Answer Existence Judgment tasks.

We construct Mid-Level tasks by reformulating the original MCQs in MedQA and MedMCQA into statement verification tasks, multi-step rectification tasks, and answer existence judgment tasks. For the Statement Validation task, we prompt GPT-4o to generate a statement based on the question and a given option, using the following prompt format:

*You are a medical expert and you are given a multiple choice question. Please change the question into a statement verification based on a given option.*

*Example 1:*

***Input***

*Original Question: Gene for Dentin mineralization*

*Options: A: MAP1B B: PHEX C: DEN D: PHIX*

*Answer: B*

*Given Option: A*

***Output***

*Statement: MAP1B is the gene for Dentin mineralization.*

*Label: False*

*Example 2:*

***Input***

*Original Question: Child of Vasanthi was weaned from breast milk on the 5th day and was given sugarcane juice the child developed hypoglycemia and hepatomegaly biochemical examination showed hypophosphatemia and enzyme deficiencies-reducing substances in urine. The child is probably suffering from which of the following enzyme deficiencies -*

*Options: A: Fructokinase B: Aldolase B C: Glucose 6 Phosphatase D: Beta galactosidase*

*Answer: B*

*Given Option: B*

***Output***

*Statement: Child of Vasanthi was weaned from breast milk on the 5th day and was given sugarcane juice the child developed hypoglycemia and hepatomegaly biochemical examination showed hypophosphatemia and enzyme deficiencies-reducing substances in urine. The child is probably suffering from the deficiency of enzyme Aldolase B.*

*Label: True*

*Test:*

***Input***

*Original Question: [Original MCQ Question]*

*Options: [Options]*

*Answer: [Correct Answer]*

*Given Option: [Give Option]*

***Output***

We found that providing both the correct answer and the given option in the prompt can help the model generate more accurate statements. We also provide two examples of the generation process in the prompt to help the model understand the task. For the Answer Existence Judgment task, we prompt GPT-4o to generate a distractor based on the question and the correct answer:

> *The following is a medical question along with its correct option. To increase the difficulty of the question, please generate a misleading distractor based on the correct option. This distractor should be an incorrect answer to the question. Directly generate the distractor without extra content.*
>
> *Question: [Original MCQ Question]*
>
> *Options: [Options]*
>
> *Correct Option: [Correct Answer]*
>
> *Distractor:*

To ensure that GPT-4o can correctly generate the desire statements and distractors, we have three experienced medical experts manually verify a batch of 100 generated statements and distractors. We found that GPT-4o can generate correct statements and distractors with an accuracy of around 95%.

For generating questions of Mid-Level, we leverage question templates illustrated in Figure 10 to generate corresponding questions, using options and correct answers from the original MCQs, the generated statements, and the generated distractors. For evaluation, we use the same five-shot in-context learning setting as in Low-Level tasks.

### A.3. High-Level Tasks

For the High-Level task, following Hager et al. (2024), we construct a full-path clinical-diagnosis evaluation dataset based on the MIMIC-IV dataset. MIMIC-IV is a large-scale, de-identified, and publicly available dataset that contains electronic health records (EHRs) of patients admitted in Beth Israel Deaconess Medical Center in Boston, MA. To make the generated dataset align with tasks in the previous levels, we first identify and extract diseases involved in the MedQA and MedMCQA datasets to form a disease pool, and then retrieve the corresponding admission records from the MIMIC-IV dataset. Specifically, we first use the ICD-10-CM codes to filter the relevant admission records for each disease, and extract the discharge diagnosis section of these admission records. To ensure that the admission records are highly relevant to the diseases, we only keep the records that the corresponding disease is in the first place of the discharge diagnosis. We further exclude the records that the discharge diagnosis contains multiple diseases in the disease pool. To ensure the statistic significance of the evaluation and balance across selected diseases, we only keep the diseases that have more than 10 admission records, and randomly sample 100 admission records for disease that have more than 100 records. As a result, we obtain 2,176 admission records for 42 diseases. The distribution of the number of admission records for each disease and their affected human body systems are shown in Figure 11.

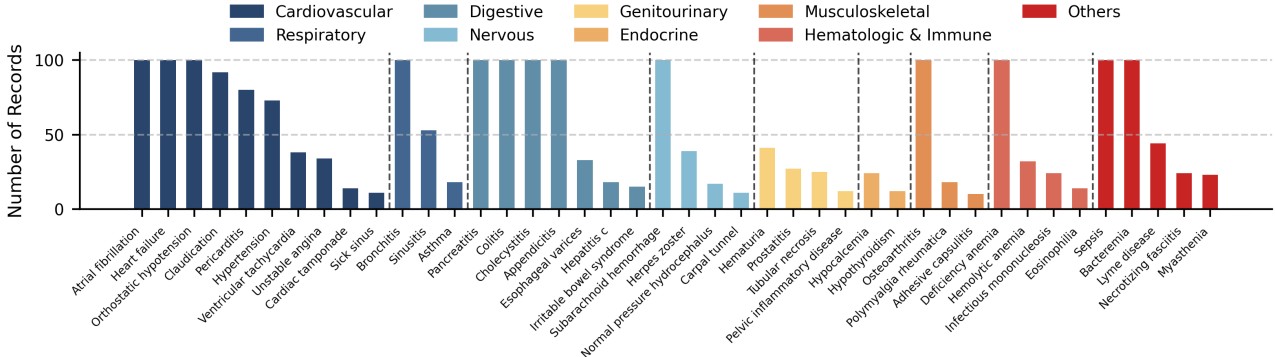

*Figure 11.* Number of admission records for each disease in the constructed full-path clinical-diagnosis evaluation dataset.

We then construct the full-path clinical-diagnosis evaluation dataset by extracting the history of present illness, physical examination sections from the clinical notes of the admission records, and the corresponding laboratory tests, microbiology

tests, and imaging tests from the clinical data tables. Then, we construct an agent-based evaluation setting, where the model is required to make decisions (e.g., request specific tests, make the final diagnosis) based on the information it obtain through the interaction process. Specifically, for each admission record, we first provide the history of present illness to the model, using the following prompt format:

> *You are an experienced medical AI assistant. Your ultimate goal is to help the doctor diagnose the patient's condition. You will be provided with the patient's history and the results of any tests that the doctor has already performed. You can also order additional tests for more information, including physical examinations, laboratory tests, microbiology tests, and imaging.*
>
> *The action you can choose are:*
>
> *1. PE: Perform physical examination of patient and receive the observations.*
>
> *2. LAB: Run laboratory tests and receive their values. You will get all the lab tests results at once.*
>
> *3. MICRO: Run microbiology tests and receive their values. You will get all the microbiology tests results at once.*
>
> *4. IMAGE: Do specific imaging scans and receive the radiologist report. You will get all the imaging results at once.*
>
> *5. OUTPUT: Output the final diagnosis.*
>
> *Note: To improve diagnostic efficiency, please perform physical examinations (PE), laboratory tests (LAB), microbiological tests (MICRO), and imaging scans (IMAGE) only when necessary for diagnosis. When you are confident, choose the "OUTPUT" action and you will be asked to output the corresponding diagnosis.*
>
> *Your output format should be:*
>
> *Rationale: (your reasoning process for choosing the next action)*
>
> *Action: (one of the actions in [PE, LAB, IMAGE, MICRO, OUTPUT])*
>
> *Now a patient comes to see the doctor.*
>
> *Patient History: [History of Present Illness].*
>
> *Please choose your next action from [PE, LAB, IMAGE, MICRO, OUTPUT].*

If the model chooses the "PE" action, we provide the physical examination section of the clinical notes to the model using the following prompt format:

> *Physical Examination of this patient: [Physical Examination].*
>
> *Please choose your next action from [Lefted Actions].*

Otherwise, if the model chooses one of the "LAB", "MICRO", or "IMAGE" actions, we will further ask the model to provide the detailed list of examination items of the corresponding test type:

> *You choose [Model's Action] as the next action. Please provide the specific list of [Model's Action] you want to run:*
>
> *please output your choice in the following format:*
>
> *Rationale: (your reasoning process for choosing the next action)*
>
> *Lab Tests: (the specific laboratory tests you want to run, separated by commas)*

To parse the model's requested examination items, we leverage the name of examination items appeared in MIMIC-IV to form an examination pool. Then, we use UMLS to map the examination items in the pool to the corresponding CUIs, and retrieve the corresponding synonyms of the CUIs. Finally, we extract and match the synonyms of the CUIs with the model's requested examination items using fuzzy matching. If the similarity between the synonyms and the model's requested examination items is above a certain threshold (0.9), we consider the examination items are correctly parsed. We further conduct manual verification over the examination items that are not identified by the fuzzy matching, and construct a mapping table between the missing examination items and their standard names in MIMIC-IV. Finally, we verify the recall of the examination items parsing process by randomly sampled 100 recognition results randomly sampled from the evaluated models' outputs, and have three experienced doctors manually verify the results. We calculate the recall of the examination items parsing process by comparing the manual verification results with the recognition results of the evaluated models. We find that the recall is around 95% across all three types of tests.

After the parsing process, we provide the model with the results of the requested tests, and ask the model to choose the next action:

*Here are the [Model's Action] results of this patient:*

*[Results of the requested tests]*

*Please choose your next action from [Lefted Actions].*

If the model fails to output the valid action, we will warn the model and ask the model to choose the next action again:

*You have chosen an action that is not available or used the wrong format.*

*Your output format should be:*

*Rationale: (your reasoning process for choosing the next action)*

*Action: (one of the actions in [Lefted Actions])*

*Please choose your next action from [Lefted Actions].*

Once the model chooses the "OUTPUT" action, we will further ask the model to output the final diagnosis:

*You choose "OUTPUT" as the next action. Please output the final diagnosis for this patient.*

*Your output format should be:*

*Diagnosis: (the final diagnosis, specific disease name)*

We use the zero-shot learning setting to evaluate the performance of LLMs on this full-path clinical-diagnosis task. When the chat history length exceed the maximum token length of the model, we ask the model to summarize the chat history and provide the summary to the model as the input prompt:

*Summarize our chat history, condense the content as much as possible while preserving all essential information related to the diagnosis. Eliminate redundant or irrelevant information, and ensure the summary maintains coherence and clarity.*

*Your output format should be:*

*Summary: (your summary of the dialogue)*

For answer parsing, we construct a disease synonyms mapping based the UMLS matching and manual verification process, and use the mapping to match the model's output with the ground truth. We also record models' requested examination items to calculate the full-path clinical-diagnosis accuracy.

## B. Details of Evaluation Metrics

As introduced in Section 3.3, we use accuracy as the evaluation metric for Low-Level and Mid-Level tasks. For High-Level tasks, we use the full-path clinical-diagnosis accuracy as the evaluation metric. To make the evaluation metrics across different levels comparable, we normalize the accuracy of Low-Level and Mid-Level tasks using Equation (4). Specifically, for the Low-Level tasks (MCQs), we set the random accuracy as $\frac{1}{N_c}$, where $N_c$ is the number of options in the MCQs. For the Statement Validation tasks and Answer Existence Judgment tasks, we set the random accuracy as 0.5, as there are only two possible labels. For the Multi-step Rectification tasks, we find that the random accuracy of questions with the correct answer provided is much higher than that of questions with the wrong answer provided, because for the latter case, the model need to further choose the correct answer from the rest of $N_c - 1$ options. Therefore, we consider weight the accuracy of the questions with the correct answer provided by $\alpha$ and the accuracy of the questions with the wrong answer provided by $1 - \alpha$ to make the random accuracy independent of the random strategy. Considering a random strategy that deciding the given option is correct with a probability of $p$, the random accuracy of the Multi-step Rectification tasks is calculated as:

$$\text{Accuracy}_{rand} = \alpha \times p + (1 - \alpha) \times (1 - p) \times \frac{1}{N_c - 1}$$

Since we want the random accuracy be invariant to the random strategy, the derivative of the random accuracy with respect to $p$ should be zero, which leads to $\alpha = \frac{1}{N_c}$. Therefore, we set $\alpha = \frac{1}{N_c}$ for the Multi-step Rectification tasks, and the random accuracy is $\frac{1}{N_c}$ as well.

For the full-path clinical-diagnosis accuracy, since the task format is open-ended, we set the random accuracy as zero and directly report the accuracy of the evaluated models.

## C. Details of Evaluated Models

We list the basic information of LLMs evaluated in our study in Table 7. The models are divided into general and medical-domain specific models. We denote Phi3-mini-Instruct-128k, Phi3-small-Instruct-128k, Phi3-medium-Instruct-128k, and Phi4 as Phi3-3.8B, Phi3-7B, Phi3-14B, and Phi4-14B, for simplicity.

*Table 7.* The basic information of large language models evaluated in our study, including the model type, family, backbone model, parameter size, and training data size.

| Model Type | Family | Model | Backbone Model | Parameter Size (B) | Training Data Size (T) |
|---|---|---|---|---|---|
| General Domain | Llama | Llama-7B | - | 7 | 1.4 |
| | | Llama-13B | - | 13 | 1.4 |
| | | Llama-33B | - | 33 | 1.4 |
| | | Llama-65B | - | 65 | 1.4 |
| | | Llama2-7B | - | 7 | 2 |
| | | Llama2-13B | - | 13 | 2 |
| | | Llama2-70B | - | 70 | 2 |
| | | Llama3-8B | - | 8 | 15 |
| | | Llama3-70B | - | 70 | 15 |
| | Qwen | Qwen-7B | - | 7 | 3 |
| | | Qwen-14B | - | 14 | 3 |
| | | Qwen-72B | - | 72 | 3 |
| | | Qwen2-7B | - | 7 | 7 |
| | | Qwen2-72B | - | 72 | 7 |
| | | Qwen2.5-7B | - | 7 | 18 |
| | | Qwen2.5-14B | - | 14 | 18 |
| | | Qwen2.5-32B | - | 32 | 18 |
| | | Qwen2.5-72B | - | 72 | 18 |
| | Gemma | Gemma-2B | - | 2 | 3 |
| | | Gemma-7B | - | 7 | 6 |
| | | Gemma2-2B | - | 2 | 2 |
| | | Gemma2-9B | - | 9 | 8 |
| | | Gemma2-27B | - | 27 | 13 |
| | Phi | Phi3-3.8B | - | 3.8 | 3.3 |
| | | Phi3-7B | - | 7 | 4.8 |
| | | Phi3-14B | - | 14 | 3.8 |
| | | Phi4-14B | - | 14 | 9.8 |
| | GPT | GPT-4o-mini | - | Not Available | Not Available |
| | | GPT-4o | - | Not Available | Not Available |
| | | o3-mini | - | Not Available | Not Available |
| | DeepSeek | DeepSeek-V3 | - | 671 (37 activated) | 14.8 |
| | | DeepSeek-R1 | - | 671 (37 activated) | 14.8 |
| Medical Domain | Others | ClinicalCamel-70B | Llama2-70B | 70 | 2 |
| | | Med42-70B | Llama2-70B | 70 | 2 |
| | | Meditron-70B | Llama2-70B | 70 | 2 |
| | | Med42-v2-8B | Llama3-8B | 8 | 15 |
| | | Med42-v2-70B | Llama3-70B | 70 | 15 |
| | | Meditron3-8B | Llama3-8B | 8 | 15 |
| | | Meditron3-70B | Llama3-70B | 70 | 15 |
| | | MMed-Llama3-8B | Llama3-8B | 8 | 15 |

## D. Details of Evaluation Settings

For the Low-Level and Mid-Level tasks, we set the model temperature as 0 to ensure the model outputs the most confident answer. We conduct five repeated experiments by randomly selecting five demonstrative samples from the rest of dataset. We then calculate the average accuracy of the five experiments, and report the mean and standard error of the accuracy across all test samples. For the High-Level tasks, considering that low temperature may lead to the model being too conservative, we set the model temperature as 0.8 to encourage the model to explore more possibilities. We also verified through preliminary experiments that LLMs produce stable outputs with this parameter setting. We conduct repeated experiments three times and report the mean and standard error of the accuracy.

# E. Details of Evaluation Results

We list in Table 8 the detailed results of the evaluated LLMs on tasks across all cognitive levels. The results are reported in the format of mean and standard error of the accuracy. We notice that some medical models (Meditron-70B, MMed-Llama3-8B) achieve a near-to-zero performance on the High-Level tasks. After carefully checking the outputs of these models, we find that these models fail to follow the instruction and tend to repeat the given instruction or output irrelevant information.

*Table 8.* Full performance (mean and standard error of normalized performance (%)) of Large Language Models on tasks across different cognitive levels. Considering the high cost of GPT-4o, o3-mini, DeepSeek-V3, and DeepSeek-R1, we only evaluate these models with three repeated experiments.

| Model | Low-Level | | Mid-Level | | | | | | High-Level |
| | | | StmtVal | | AnsExist | | MultiRect | | |
| | MedQA | MedMCQA | MedQA | MedMCQA | MedQA | MedMCQA | MedQA | MedMCQA | MIMIC-IV |
|---|---|---|---|---|---|---|---|---|---|
| Llama-7B | $3.36 \pm 0.82$ | $10.40 \pm 0.24$ | $4.88 \pm 0.84$ | $4.52 \pm 0.45$ | $0.86 \pm 0.70$ | $0.55 \pm 0.43$ | $1.60 \pm 0.63$ | $8.06 \pm 0.26$ | - |
| Llama-13B | $13.18 \pm 0.57$ | $18.98 \pm 0.22$ | $6.47 \pm 0.74$ | $6.80 \pm 0.57$ | $-0.05 \pm 0.63$ | $0.49 \pm 0.54$ | $1.45 \pm 0.26$ | $2.17 \pm 0.16$ | - |
| Llama-33B | $24.09 \pm 0.26$ | $26.88 \pm 0.34$ | $16.70 \pm 0.24$ | $12.29 \pm 0.48$ | $1.86 \pm 0.26$ | $1.03 \pm 0.27$ | $19.91 \pm 0.84$ | $17.99 \pm 0.34$ | - |
| Llama-65B | $26.64 \pm 0.49$ | $29.59 \pm 0.29$ | $22.26 \pm 0.65$ | $17.03 \pm 0.51$ | $5.53 \pm 1.01$ | $2.59 \pm 0.20$ | $22.60 \pm 0.31$ | $20.83 \pm 0.18$ | - |
| Llama2-7B | $15.88 \pm 0.28$ | $18.32 \pm 0.22$ | $8.55 \pm 0.70$ | $7.15 \pm 0.59$ | $0.25 \pm 1.29$ | $0.47 \pm 0.22$ | $3.88 \pm 0.43$ | $8.58 \pm 0.31$ | $2.65 \pm 0.20$ |
| Llama2-13B | $21.10 \pm 0.30$ | $20.94 \pm 0.33$ | $9.58 \pm 1.77$ | $12.44 \pm 0.32$ | $1.08 \pm 0.92$ | $1.25 \pm 0.35$ | $7.84 \pm 0.32$ | $8.65 \pm 0.22$ | $4.92 \pm 0.15$ |
| Llama2-70B | $37.99 \pm 0.34$ | $35.60 \pm 0.26$ | $23.77 \pm 0.43$ | $17.14 \pm 0.17$ | $5.51 \pm 0.76$ | $2.58 \pm 0.68$ | $33.47 \pm 0.52$ | $28.65 \pm 0.26$ | $5.16 \pm 0.18$ |
| Llama3-8B | $37.30 \pm 0.53$ | $39.11 \pm 0.29$ | $22.77 \pm 0.56$ | $22.10 \pm 0.31$ | $4.83 \pm 0.82$ | $3.81 \pm 0.25$ | $6.80 \pm 0.19$ | $8.29 \pm 0.15$ | $10.50 \pm 0.16$ |
| Llama3-70B | $63.68 \pm 0.34$ | $60.82 \pm 0.28$ | $53.16 \pm 0.46$ | $41.29 \pm 0.34$ | $9.36 \pm 1.01$ | $10.37 \pm 0.64$ | $62.87 \pm 0.29$ | $57.07 \pm 0.16$ | $17.81 \pm 0.41$ |
| Qwen-7B | $19.40 \pm 0.31$ | $24.85 \pm 0.21$ | $14.19 \pm 0.49$ | $13.00 \pm 0.69$ | $2.74 \pm 0.93$ | $1.00 \pm 0.25$ | $5.09 \pm 0.43$ | $7.24 \pm 0.21$ | $2.51 \pm 0.31$ |
| Qwen-14B | $38.05 \pm 0.38$ | $39.25 \pm 0.13$ | $21.13 \pm 0.30$ | $18.56 \pm 0.32$ | $4.93 \pm 1.21$ | $3.06 \pm 0.27$ | $27.89 \pm 0.09$ | $26.31 \pm 0.22$ | $4.14 \pm 0.29$ |
| Qwen-72B | $50.60 \pm 0.39$ | $50.95 \pm 0.18$ | $36.43 \pm 0.66$ | $30.99 \pm 0.36$ | $10.42 \pm 0.55$ | $7.09 \pm 0.18$ | $38.26 \pm 0.41$ | $40.74 \pm 0.49$ | $5.89 \pm 0.35$ |
| Qwen2-7B | $36.73 \pm 0.44$ | $39.85 \pm 0.15$ | $28.08 \pm 0.20$ | $24.66 \pm 0.30$ | $4.00 \pm 0.76$ | $4.68 \pm 0.25$ | $31.60 \pm 0.43$ | $33.46 \pm 0.24$ | $6.99 \pm 0.25$ |
| Qwen2-72B | $65.91 \pm 0.24$ | $60.41 \pm 0.19$ | $55.09 \pm 0.23$ | $42.23 \pm 0.23$ | $24.00 \pm 0.58$ | $11.67 \pm 0.14$ | $62.86 \pm 0.27$ | $57.68 \pm 0.26$ | $15.10 \pm 0.09$ |
| Qwen2.5-7B | $42.86 \pm 0.49$ | $45.39 \pm 0.17$ | $30.29 \pm 0.34$ | $26.43 \pm 0.38$ | $13.66 \pm 0.23$ | $7.49 \pm 0.32$ | $37.08 \pm 0.22$ | $40.34 \pm 0.19$ | $9.50 \pm 0.04$ |
| Qwen2.5-14B | $52.23 \pm 0.30$ | $51.74 \pm 0.24$ | $40.86 \pm 0.49$ | $33.98 \pm 0.29$ | $17.36 \pm 0.56$ | $11.82 \pm 0.50$ | $48.67 \pm 0.34$ | $46.40 \pm 0.15$ | $12.07 \pm 0.21$ |
| Qwen2.5-32B | $59.21 \pm 0.21$ | $57.17 \pm 0.09$ | $48.05 \pm 0.43$ | $37.92 \pm 0.06$ | $14.31 \pm 0.60$ | $11.36 \pm 0.36$ | $53.01 \pm 0.33$ | $51.04 \pm 0.11$ | $11.87 \pm 0.22$ |
| Qwen2.5-72B | $67.83 \pm 0.20$ | $61.82 \pm 0.12$ | $55.70 \pm 0.42$ | $44.79 \pm 0.53$ | $28.98 \pm 0.65$ | $18.54 \pm 0.39$ | $63.70 \pm 0.36$ | $58.39 \pm 0.13$ | $16.05 \pm 0.24$ |
| Gemma-2B | $6.16 \pm 0.63$ | $15.04 \pm 0.24$ | $1.21 \pm 0.41$ | $4.12 \pm 0.28$ | $-1.28 \pm 1.01$ | $-0.34 \pm 0.41$ | $1.67 \pm 0.49$ | $4.72 \pm 0.32$ | $0.59 \pm 0.09$ |
| Gemma-7B | $30.35 \pm 0.42$ | $27.41 \pm 0.38$ | $17.03 \pm 0.49$ | $18.90 \pm 0.10$ | $5.16 \pm 0.34$ | $2.89 \pm 0.31$ | $23.05 \pm 0.32$ | $19.14 \pm 0.43$ | $0.33 \pm 0.09$ |
| Gemma2-2B | $14.87 \pm 0.34$ | $24.18 \pm 0.31$ | $4.96 \pm 0.62$ | $10.23 \pm 0.45$ | $-0.48 \pm 0.58$ | $0.68 \pm 0.69$ | $0.86 \pm 0.30$ | $0.63 \pm 0.05$ | $5.02 \pm 0.33$ |
| Gemma2-9B | $44.21 \pm 0.18$ | $46.20 \pm 0.20$ | $32.86 \pm 0.57$ | $27.27 \pm 0.31$ | $6.97 \pm 0.74$ | $6.76 \pm 0.40$ | $33.78 \pm 0.32$ | $35.11 \pm 0.32$ | $9.04 \pm 0.28$ |
| Gemma2-27B | $53.65 \pm 0.31$ | $50.45 \pm 0.18$ | $41.36 \pm 0.68$ | $35.66 \pm 0.22$ | $10.59 \pm 1.17$ | $10.14 \pm 0.33$ | $48.23 \pm 0.16$ | $45.18 \pm 0.25$ | $11.19 \pm 0.64$ |
| Phi3-3.8B | $37.52 \pm 0.28$ | $40.96 \pm 0.20$ | $29.01 \pm 0.72$ | $24.40 \pm 0.04$ | $10.01 \pm 1.19$ | $7.50 \pm 0.29$ | $30.31 \pm 0.54$ | $35.70 \pm 0.40$ | $2.21 \pm 0.28$ |
| Phi3-7B | $49.25 \pm 0.30$ | $46.95 \pm 0.30$ | $38.92 \pm 0.29$ | $27.63 \pm 0.25$ | $4.83 \pm 0.45$ | $2.13 \pm 0.23$ | $42.61 \pm 0.24$ | $35.58 \pm 0.30$ | $7.74 \pm 0.16$ |
| Phi3-14B | $54.06 \pm 0.49$ | $49.70 \pm 0.27$ | $40.45 \pm 0.59$ | $31.92 \pm 0.26$ | $5.79 \pm 0.52$ | $8.24 \pm 0.48$ | $48.75 \pm 0.46$ | $45.21 \pm 0.20$ | $3.89 \pm 0.01$ |
| Phi4-14B | $56.19 \pm 0.40$ | $52.62 \pm 0.30$ | $44.88 \pm 0.55$ | $35.28 \pm 0.25$ | $19.92 \pm 1.04$ | $15.13 \pm 0.23$ | $53.16 \pm 0.26$ | $49.98 \pm 0.15$ | $12.18 \pm 0.37$ |
| GPT-4o-mini | $62.70 \pm 0.17$ | $55.40 \pm 0.10$ | $47.97 \pm 0.44$ | $36.46 \pm 0.59$ | $24.98 \pm 0.74$ | $12.32 \pm 0.57$ | $56.11 \pm 0.51$ | $48.12 \pm 0.19$ | $13.81 \pm 0.84$ |
| GPT-4o | $78.33 \pm 0.75$ | $64.00 \pm 1.02$ | $67.83 \pm 0.60$ | $49.67 \pm 0.33$ | $33.33 \pm 2.20$ | $11.83 \pm 2.91$ | $70.92 \pm 1.23$ | $63.39 \pm 0.39$ | $19.33 \pm 0.30$ |
| o3-mini | $87.71 \pm 0.55$ | $\mathbf{74.44 \pm 0.80}$ | $76.67 \pm 0.17$ | $57.50 \pm 0.76$ | $\mathbf{62.83 \pm 0.93}$ | $30.50 \pm 1.89$ | $86.17 \pm 0.91$ | $73.56 \pm 1.12$ | $15.05 \pm 0.45$ |
| DeepSeek-V3 | $78.33 \pm 0.75$ | $65.78 \pm 0.22$ | $56.50 \pm 0.63$ | $50.83 \pm 1.17$ | $5.17 \pm 0.17$ | $7.33 \pm 1.69$ | $72.71 \pm 0.40$ | $62.06 \pm 0.11$ | $19.42 \pm 0.17$ |
| DeepSeek-R1 | $\mathbf{89.79 \pm 0.75}$ | $\mathbf{74.44 \pm 0.80}$ | $\mathbf{77.33 \pm 1.33}$ | $\mathbf{62.00 \pm 0.29}$ | $54.17 \pm 1.33$ | $\mathbf{36.33 \pm 1.09}$ | $\mathbf{87.67 \pm 0.65}$ | $\mathbf{75.44 \pm 0.11}$ | $\mathbf{26.54 \pm 0.31}$ |
| ClinicalCamel-70B | $42.20 \pm 0.52$ | $37.14 \pm 0.34$ | $29.48 \pm 0.32$ | $21.68 \pm 0.22$ | $7.09 \pm 1.02$ | $4.82 \pm 0.32$ | $36.82 \pm 0.63$ | $32.80 \pm 0.17$ | $2.19 \pm 0.08$ |
| Med42-70B | $49.18 \pm 0.28$ | $48.11 \pm 0.21$ | $33.41 \pm 0.72$ | $21.74 \pm 0.48$ | $12.65 \pm 0.54$ | $7.37 \pm 0.56$ | $40.01 \pm 0.78$ | $36.77 \pm 0.24$ | $4.25 \pm 0.28$ |
| Meditron-70B | $42.52 \pm 0.59$ | $36.18 \pm 0.31$ | $28.15 \pm 0.65$ | $20.89 \pm 0.47$ | $8.18 \pm 1.05$ | $4.23 \pm 0.32$ | $37.64 \pm 0.33$ | $29.80 \pm 0.36$ | $0.32 \pm 0.01$ |
| Llama3-Med42-8B | $46.42 \pm 0.64$ | $47.13 \pm 0.18$ | $32.10 \pm 0.65$ | $26.31 \pm 0.12$ | $9.31 \pm 0.60$ | $7.08 \pm 0.35$ | $40.47 \pm 0.29$ | $41.81 \pm 0.25$ | $11.22 \pm 0.21$ |
| Llama3-Med42-70B | $65.88 \pm 0.12$ | $62.74 \pm 0.19$ | $56.23 \pm 0.29$ | $44.08 \pm 0.24$ | $20.86 \pm 1.13$ | $15.39 \pm 0.21$ | $62.68 \pm 0.28$ | $58.83 \pm 0.17$ | $12.37 \pm 0.18$ |
| Meditron3-8B | $40.60 \pm 0.26$ | $45.37 \pm 0.20$ | $24.10 \pm 0.23$ | $23.13 \pm 0.36$ | $4.48 \pm 0.71$ | $5.71 \pm 0.34$ | $25.28 \pm 0.37$ | $27.45 \pm 0.27$ | $7.61 \pm 0.49$ |
| Meditron3-70B | $68.96 \pm 0.27$ | $62.27 \pm 0.17$ | $54.82 \pm 0.29$ | $41.17 \pm 0.29$ | $6.52 \pm 1.22$ | $9.57 \pm 0.55$ | $61.70 \pm 0.52$ | $54.76 \pm 0.18$ | $10.36 \pm 0.10$ |
| MMed-Llama3-8B | $37.96 \pm 0.43$ | $39.20 \pm 0.21$ | $10.34 \pm 0.39$ | $21.78 \pm 0.55$ | $5.31 \pm 0.27$ | $5.82 \pm 0.33$ | $24.64 \pm 0.24$ | $24.60 \pm 0.16$ | $0.32 \pm 0.01$ |

# F. Details of Clinician Validation

To validate that the cognitive difficulties of tasks in our benchmark are consistent with the cognitive levels defined in Section 3.2, we conducted a clinician validation. Specifically, we first randomly sampled 20 questions from each task to form a subset of 100 questions. For Low-Level and Mid-Level tasks, we sample questions from the MedQA benchmark. Then, we recruited four licensed clinicians with 3 to 8 years of experience to evaluate this subset from two perspectives: (1) their accuracy in answering the questions and (2) their subjective difficulty ratings to the questions (1=Easy, 10=Hard). The detailed labeling instructions are as follows:

> Please answer the following questions based on your medical knowledge. For each question, please provide your answer and rate the difficulty of the question on a scale from 1 to 10, where 1 means "very easy" and 10 means "very hard".
>
> - **Low-Level Tasks**: These tasks require basic medical knowledge and understanding of medical concepts. Please choose the most appropriate answer from the given options.
> - **Mid-Level Tasks**: These tasks require a more complex application of medical knowledge. For statement validation tasks, please determine whether the statement is true or false. For answer existence judgment tasks, please determine whether the answer exists in the given options. For multi-step rectification tasks, please first determine whether the provided answer is correct, and if not, provide the correct answer.
> - **High-Level Tasks:** Given the history of present illness section from an inpatient admission note, the task is to provide the patient's primary diagnosis.
>   At each step, the following operation types are allowed:
>   (1) Physical Examination: Request a physical examination for the patient;
>   (2) Laboratory Tests: Request laboratory tests for the patient;
>   (3) Microbiological Culture: Request microbiological cultures (e.g., blood, urine);
>   (4) Imaging: Request imaging tests (e.g., CT, X-ray, ultrasound);
>   (5) Diagnosis: A diagnosis should be provided when (a) the test results are sufficient to support a diagnosis, or (b) all possible tests have been completed. Output the patient's primary diagnosis (can include multiple diagnoses), including any conditions already mentioned in the HPI.
>   **Note**: Please record the sequence in which the test types were requested for diagnosis.

