# OpenReview forum: "Evaluating LLMs Across Multi-Cognitive Levels: From Medical Knowledge Mastery to Scenario-Based Problem Solving"
_ICML.cc/2025/Conference — ICML 2025 poster_

### Official Review · Reviewer_VJ98 · 2025-02-28

**Overall Recommendation:** 3

**Summary:**

This paper presents an evaluation framework inspired by Bloom’s Taxonomy, integrating multiple tasks to reflect diverse cognitive levels. The author evaluates popular general and medical LLMs, observing a significant performance decline in performance with the increasing of cognitive complexity.
Their findings highlight the importance of scaling up models' parameter sizes to more effectively tackle clinical challenges.

## update after rebuttal
In the rebuttal stage, the authors provide details about the human evaluation of the cognitive levels. These evidence addressed my concerns regarding the experiment design. The paper proposed a valuabale evaluation pipeline to analysis the clinical ultility of LLMs.
Therefore, I update my score to 3 (weak accept).
However, I still think the findings within this paper, larger LLMs performs better in more difficult tasks, is not supervising. I hope the author could provide more in-depth analysis regrading these results.

**Claims And Evidence:**

Problematic claim:
1. The author claims that question-answering (QA) is a task with low cognitive complexity, as large language models (LLMs) primarily need to memorize the medical information. However, some clinical questions in MedQA[1] require that LLMs analyze symptoms and engage in intricate reasoning to arrive at the correct answer.

2. The cognitive complexity of Mid-Level tasks (such as statement validation questions) is influenced by the question's complexity. An example from the MedMCQA[2] dataset illustrates this:
"Q: Which vitamin is provided solely by animal sources?" represents a straightforward knowledge question. Even if reformulated into a statement validation question such as: "Does Vitamin C come exclusively from animal sources?", it remains a simple task, requiring the LLM simply to memorize relevant information, without applying it to clinical scenarios.

In summary, the author classifies different cognitive levels based on the tasks with different input and output formats. However, in my opinion, a more suitable way to classify cognitive levels is based on the complexity of questions.

[1] Jin, Di, et al. "What disease does this patient have? a large-scale open domain question answering dataset from medical exams." Applied Sciences 11.14 (2021): 6421.

[2] Pal, Ankit, Logesh Kumar Umapathi, and Malaikannan Sankarasubbu. "Medmcqa: A large-scale multi-subject multi-choice dataset for medical domain question answering." Conference on health, inference, and learning. PMLR, 2022.

**Essential References Not Discussed:**

The key contribution of this paper is evaluating LLMs from different cognitive levels. However, previous works [4][5] have thoroughly evaluated LLMs across a variety of tasks and dimensions. It is important for the authors to discuss these studies.

[4] Wu, Chaoyi, et al. "Towards evaluating and building versatile large language models for medicine." npj Digital Medicine 8.1 (2025): 58.

[5] Johri, Shreya, et al. "An evaluation framework for clinical use of large language models in patient interaction tasks." Nature Medicine (2025): 1-10.

**Experimental Designs Or Analyses:**

1. The evaluation of different LLMs on the proposed benchmark ignores state-of-the-art reasoning LLMs (such as openai o1 and o3-mini, Deepseek-R1).

2. This paper ignores the evaluation of LLMs from (1) post-training (2) inference-time scaling dimensions. How different post-training and inference-time strategies can enhance the capability of LLMs at different cognitive levels is not explored.

**Methods And Evaluation Criteria:**

This paper adopts accuracy and full-path diagnosis accuracy as evaluation metrics. I think this part is reasonable.

**Other Comments Or Suggestions:**

I will increase my overall score if the authors can address my concerns mentioned above.

**Other Strengths And Weaknesses:**

Strength: this paper comprehensively evaluates a wide range of popular LLMs across different tasks, which is valuable for the community to better understand the clinical capability of LLMs.

Weakness: the classification of different cognitive levels is not convincing enough for me. Furthermore, this paper offers limited insights regarding how to enhance the performance of medical LLMs on high cognitive level tasks beyond merely increasing parameter sizes.

**Questions For Authors:**

Please refer to previous 'Claims and Evidence', 'Method and Evaluation Criteria', 'Experimental Designs or Analyses', and 'Other Strengths and Weaknesses' parts.

**Relation To Broader Scientific Literature:**

The findings within this paper are aligned with the scaling law[3].

[3] Kaplan, Jared, et al. "Scaling laws for neural language models." arXiv preprint arXiv:2001.08361 (2020).

**Theoretical Claims:**

This paper does not contain theoretical claims.

---

> ### Author Rebuttal · Authors · 2025-04-01
>
> We sincerely appreciate your thoughtful and constructive feedback. Below, we address each of the concerns you raised.
>
> 1. **Benchmark design issue** : We highly appreciate your kind comments. Indeed, the cognitive difficulty of a question is affected by multiple factors, including task type and question complexity. However, the difficulty of evaluation questions is typically constrained by the task format. For example, while the MedQA dataset is designed to evaluate LLMs’ ability to analyze clinical cases, its multiple-choice format somewhat limits its ability to fully reflect how well LLMs request and integrate diagnostic information in real clinical settings. In fact, the task formats in our benchmark inherently reflects variations in cognitive difficulty: Mid-level tasks reflect the impact of increasing reasoning steps and expanding decision space on performance, while high-level tasks further evaluating LLMs in requesting and integrating critical information for decision-making in clinical scenarios. While there is indeed some variation in difficulty among questions within the same task format, the constructed benchmark generally meets our design expectations.
>
>    We also conducted a clinician evaluation to validate the proposed benchmark. Specifically, we first randomly sampled 20 questions from each task to form a subset of 100 questions. Then, we recruited four licensed clinicians with 3 to 8 years of experience to evaluate this subset from two perspectives: (1) their accuracy in answering the questions and (2) their subjective difficulty ratings to the questions (1=Easy, 10=Hard). The evaluation results are as follows:
>
>    |Cognitive Level|Clinician Accuracy|Clinician Subjective Difficulty|
>    |-|:-:|:-:|
>    |Low|68.8|Mean:5.0 Median: 5.5|
>    |Mid|54.2|Mean:6.0 Median: 5.8|
>    |High|23.5|Mean:7.5 Median: 7.5|
>
>    The experimental results show that clinicians’ accuracy decreases as the cognitive level increases, aligning with their subjective difficulty rating. This indicates that the designed tasks successfully achieved the intended difficulty levels and can effectively reflect how far LLMs are from effectively solving real clinical problems. Thank you again for your kind suggestions, and we will add this clinician evaluation into the revised paper.
>
> 2. **Evaluation of reasoning LLMs**: Thank you for your constructive suggestions. Indeed, since our paper submission, reasoning LLMs have been receiving increasing attention. Following your kind suggestions, we primarily evaluated two typical reasoning LLMs, DeepSeek-R1 and o3-mini, and compare them with SOTA chat LLMs in corresponding parameter scales. The evaluation results are as follows:
> |Model|Low-Level|Mid-Level|High-Level|
>    |-|:-:|:-:|:-:|
>    |DeepSeek-V3|78.3|44.8|19.4|
>    |**DeepSeek-R1**|**89.3**|**73.1**|**25.9**|
>    |GPT-4o-mini|62.7|43.0|13.8|
>    |**o3-mini**|**88.1**|**75.2**|**15.5**|
>
>    Results show that reasoning LLMs outperform chat LLMs across all cognitive levels, though the performance gap narrows on high-level tasks due to their significantly increased difficulty. Note that we have not evaluated other reasoning LLMs due to time and cost constraints (as the price of o1 is quite high). We plan to add this evaluation and include additional evaluations in our revised paper.
>
> 3. **Insights of medical LLM development**: Thank you for your thoughtful comments. This work provides two key insights into the development of medical LLMs:
>
>    (1) Although the performance of smaller LLMs (<10B) on medical benchmarks is gradually approaching that of larger LLMs, our study indicates that increasing parameter size remains crucial for tackling tasks in higher cognitive levels.
>
>    (2) While medical post-training (used in current medical LLMs) and inference-time scaling (applied in reasoning LLMs) strategies work on low- and mid-level tasks, effectively solving high-level tasks demand further advancements in clinical reasoning abilities—particularly in the retrieval and integration of key information for decision-making in real-world scenarios.
>
>    Again, we appreciate your comments and will further highlight our insights in the revised paper.
>
> 4. **Discussion with more benchmarks**: We are sincerely grateful for your kind suggestions. Wu et al. investigate LLMs’ ability to handle diverse medical tasks by constructing MedS-Bench, a large-scale benchmark covering 11 types of clinical tasks. Johri et al. study LLMs' ability to obtain the necessary patient information for diagnosis through diaglogue by proposing CRAFT-MD, an evaluation framework that simulates multi-turn doctor-patient interactions using a multi-agent approach. Meanwhile, our work focuses on evaluating how close LLMs are to effectively solving real-world clinical tasks by designing evaluations with progressively increasing cognitive difficulty and systematically analyzing existing LLMs. We will further enhance the discussion of other medical LLM benchmarks in our revised paper.

---

> > ### Comment · Reviewer_VJ98 · 2025-04-04
> >
> > The author's rebuttal addressed some of my concerns. But I still consists that solely modifying the MCQs does not result in tasks with different ''cognitive levels''.
> >
> > Although the author demonstrates in the rebuttal that modifying the MCQs changes the diificulty of the question, for more difficult tasks, smaller LLMs fails to scale up as good as larger LLMs is not a supervising conclusion. Therefore, I will keep my score at this stage.

---

> > > ### Author Response · Authors · 2025-04-09
> > >
> > > We sincerely appreciate your timely response.
> > >
> > > 1. **About "some of concerns" unresolved**: We truly appreciate your feedback, but frankly speaking, we are currently unsure which specific concerns you feel remain unaddressed. In our initial response, we have provided clear clarifications to each of your concern regarding (1) task settings, (2) lack of evaluation on reasoning LLMs, (3) insights into medical LLM development, and (4) comparisons with more benchmarks. Specifically, we (1) added a clinician evaluation and restated our motivation in designing tasks of different levels, (2) included evaluations of reasoning LLMs, (3) restated several key insights on how to improve the performance of medical LLMs, and (4) discussed the relations between our benchmark and other existing medical benchmarks. It's worth noting that Reviewer z2s4 has also carefully read and expressed agreement with our response to your concerns.
> > >
> > > 2. **About "solely modifying MCQs"**: We are sorry that our rebuttal did not fully explain our task settings. In fact, (1) the Mid-Level tasks are not merely modified versions of MCQs. Instead, they are carefully designed to reflect the challenges faced in real-world clinical scenarios, such as limited information and a broader decision space. (2) It is worth noting that the High-Level task is **not** derived from MCQs. Instead, they are directly constructed from electronic medical record data, aiming to assess the LLMs’ ability to actively plan, request key diagnostic information, and complete the diagnostic through reasoning.
> > >
> > > 3. **Further clarification regarding the impact of task settings on cognitive difficulty:** We feel sorry that our response did not fully convey how the task settings impact cognitive difficulty. Below, we will further explain this from several aspects: (1) We primarily constructed the Mid-Level tasks through task reformulation to keep the evaluated knowledge points unchanged (as Reviewer z2s4 mentioned, “the key knowledge point may be the same”), enhancing the comparability across tasks of different cognitive difficulties. (2) Given that Low- and Mid-Level tasks typically provide all key information at once, we further designed the High-Level task that evaluates the LLMs’ ability to actively plan and request key information based on limited patient information to complete a diagnosis.
> > >
> > >    Furthermore, for a given clinical case, providing complete patient information and limited options can largely reduce the reasoning difficulty. For example, for the following Low-Level question, LLMs can directly match the patient’s symptoms and examination results with each candidate option to easily arrive at the correct answer:
> > >
> > >    > Question: A 30-year-old woman presents to the physician because of ongoing diarrhea ... She denies any recent travel history ... Clinical examination shows mild tenderness ... Findings on colonoscopy include patchy erythema ... Mucosal biopsy shows colonic crypts ... What is the most likely diagnosis?
> > >    >
> > >    > A: Ulcerative colitis B: Crohn's Disease C: Acute infective colitis D: Pseudomembranous colitis E: Irritable bowel syndrome.
> > >
> > >    In contrast, for our High-Level tasks, where only the patient history is provided initially, LLMs are required to actively plan the next steps for examination and integrate multiple test results to reach a final diagnosis, significantly increasing the cognitive difficulty:
> > >
> > >    >A 30-year-old woman presents to the physician because of ongoing diarrhea ... She denies any recent travel history ...
> > >    >
> > >    >Your ultimate goal is to diagnose the patient's condition. You can order additional examinations for more information. Output the final diagnosis when you are confident.
> > >
> > > 4. **About LLM scaling effect**: Thank you for your comments. Our primary goal is to provide necessary insights for developing real-world usable medical LLMs through an in-depth evaluation of existing general and medical-specific LLMs. Our key findings include:
> > >
> > >    + LLMs smaller than 10B are unsuitable for higher cognitive level tasks, a conclusion with significant practical implications for real-world clinical applications. If developing a MedLLM for real-world clinical use, our findings can help guide the selection of an appropriately sized backbone model for post-training. Choosing a model that is too small may hinder achieving desired performance. In fact, our team encountered a similar challenge during the development of usable medical LLMs.
> > >
> > >    + Existing medical LLMs do not achieve significant improvements on High-Level tasks, highlighting the need to enhance LLMs' ability to actively plan, request key information, and perform reasoning based on obtained information.
> > >
> > > Once again, we sincerely appreciate your review and the concerns you raised, which led us to include the clinician evaluation and evaluations of reasoning LLMs, enhancing the completeness of our work. We also hope this response may further address your concerns unresolved.

---

### Official Review · Reviewer_kVjJ · 2025-03-12

**Overall Recommendation:** 3

**Summary:**

This paper assesses large language models (LLMs) across multiple cognitive levels, based on Bloom's taxonomy that proposes six cognitive objectives/levels in ascending order of complexity. In particular, tasks pertaining to three cognitive levels (preliminary knowledge grasp, comprehensive knowledge application, scenario-based problem solving) were defined and attempted with five state-of-the-art LLMs. It was found that LLM performance declined with increased task cognitive complexity, and that larger LLMs performed better when higher cognitive complexity was required.

## update after rebuttal
The authors' response is appreciated and improved our assessment of the study.

**Claims And Evidence:**

The claims are based on comprehensive empirical evaluation on 29 separate general-domain LLMs from five main families (Llama, Qwen, Gemma, Phi3, GPT), as well as eight medical-domain-specific LLMs (Tables 2 & 6). The main claims on performance declining with increasing task complexity as well as increasing with model size appear generally true, within each LLM model family.

**Essential References Not Discussed:**

N/A

**Experimental Designs Or Analyses:**

The experimental design appears largely sound.

**Methods And Evaluation Criteria:**

While the proposed evaluation framework (Figure 2) is reasonable, whether the (accuracy) results are directly comparable can be contested, especially between low/mid-level and high-level tasks. For example, the low-level "preliminary knowledge grasp" tasks involve MCQs with four options, the mid-level tasks involves multiple steps but fewer options (equivalence justified in Section B), and the high-level tasks appear to be free-form (Section A.3). Empirical evaluation of task difficulty by clinicians would help to calibrate/justify their appropriateness.

**Other Comments Or Suggestions:**

1. Some preliminary empirical justification for the selected model temperature (and other) parameters (as discussed in Section D) would be appropriate.

2. Some examples of LLM answers (especially for a high-level case) would be welcome, in the appendix.

**Other Strengths And Weaknesses:**

A potential weakness would be the lack of prompt engineering and chain-of-thought query techniques, that can significantly affect LLM performance.

**Questions For Authors:**

1. From Section D, it is stated that for low and mid-level tasks, the variance in reported performance metrics is due to "conduct[ing] five repeated experiments by randomly selecting five training samples from the rest of dataset". However, it is not immediately clear as to whether any additional "training" is performed for the LLMs, for purposes of this study. This statement might thus be clarified.

**Relation To Broader Scientific Literature:**

While the experiments are comprehensive, the main findings that higher-level cognitive tasks are more challenging, and that larger models fare better on such tasks, are largely unsurprising.

**Theoretical Claims:**

No theoretical claims are presented.

---

> ### Author Rebuttal · Authors · 2025-04-01
>
> We sincerely appreciate your thoughtful and constructive feedback. Below, we address each of the concerns you raised.
>
> 1. **Comparability between different cognitive levels**: Thank you for your thoughtful concern. Indeed, comparing LLMs’ performance across different cognitive levels is crucial for analyzing their medical capabilities. Considering this, we have implemented a metric alignment strategy in the "Performance Metrics" section (see Equation 4 on page 5) to eliminate interference caused by random guessing and ensure the comparability of tasks across different cognitive levels.
>
>    In fact, we have carefully considered the cognitive difficulties when designing the tasks in our benchmark. Compared to MCQs, mid-level tasks increase difficulty by removing clues, increasing reasoning steps, and expanding the decision space. High-level tasks go further by requiring LLMs to request necessary information instead of receiving it directly from the question. To ensure that the constructed benchmark meets our expectations, we followed your kind suggestions and conducted a clinician evaluation. Specifically, we randomly sampled 20 questions from each task, resulting in a clinician evaluation subset of 100 questions. Four licensed clinicians with 3 to 8 years of experience were recruited to assess the benchmark’s difficulty from two perspectives: (1) their accuracy in answering the questions and (2) their subjective difficulty ratings to the questions on a scale from 1 (Easy) to 10 (Hard). The evaluation results are as follows:
>
>    |Cognitive Level|Clinician Accuracy|Clinician Subjective Difficulty|
>    |-|:-:|:-:|
>    |Low|68.8|Mean:5.0 Median: 5.5|
>    |Mid|54.2|Mean:6.0 Median: 5.8|
>    |High|23.5|Mean:7.5 Median: 7.5|
>
>    We found that clinicians’ accuracy also decreases as the cognitive level increases across three levels. Moreover, their subjective difficulty ratings align with this trend, further demonstrating the validity of our benchmark. Once again, we appreciate your kind suggestions and will incorporate this clinician evaluation into the revised paper.
>
> 2. **LLM scale effect issue**: Thank you for your thoughtful comments. Indeed, the scaling law indicates that larger models generally perform better, but it lacks a detailed analysis of the impact of parameter size on tasks across different cognitive levels (difficulties). Considering this, we conducted a systematic evaluation to investigate parameter size effects at varying cognitive levels and found that larger LLMs significantly outperform smaller ones on harder medical tasks, demonstrating the critical role of parameter scale in real-world clinical problem-solving. Moreover, our analysis of medical-specific LLMs reveals that existing post-training methods fail to enhance high-level task performance, offering key insights for the development of medical LLMs.
>
> 3. **Evaluation setting & parameter issues**: We sincerely appreciate your thoughtful comments.
>
>    (1) Evaluation setting: For low- and mid-level tasks, we adopted the few-shot learning (widely used in LLM benchmarks), as it introduces minimal subjective bias by only guiding LLMs with examples, while prompt engineering and CoT techniques may introduce subjective biases through prompt design, potentially affecting the fairness across LLMs. For high-level tasks, regarding the reasoning difficulty, we adopted an agent-based setting [1], guiding the model to generate a rationale before producing the corresponding action.
>
>    (2) Decoding parameters: We chose the decoding parameters to align with task characteristics. For low- and mid-level tasks, we set temperature = 0 (greedy decoding) as these tasks have well-defined answer formats, and greedy decoding selects the most confident answer. For high-level tasks, we set the temperature=0.8 to balance the reasoning diversity and output stability of LLM responses. We also verified through preliminary experiments that LLMs produce stable outputs with this parameter setting.
>
>    [1] Hager P et al. Evaluation and mitigation of the limitations of large language models in clinical decision-making. Nature medicine, 2024.
>
> 4. **Examples of LLM answer**: Thank you for your constructive advice. We will include more examples of LLM answers in the appendix to better illustrate our evaluation process.
>
> 5. **Clarification of the Ambiguous Statement**: Thank you for your careful reading. We sincerely apologize for the ambiguity—here, "training samples" refer to the demonstrative examples used in few-shot in-context learning. Regarding few-shot learning, we have briefly introduced the process in Section 4.1 ("Evaluation Setting") and illustrated the corresponding input format in Figure 9. Notably, our evaluation does not involve any model training. Once again, we appreciate your careful reading. We will conduct thorough proofreading to avoid similar ambiguities in the revised manuscript.

---

### Official Review · Reviewer_z2s4 · 2025-03-14

**Overall Recommendation:** 4

**Summary:**

The paper proposed a novel medical LLM evaluation benchmark inspired by Bloom's taxonomy. Different from existing benchmarks that only evaluate the LLM on one single style of QA tasks, the proposed benchmark constructs a multi-cognitive-level evaluation framework and provides more informative results. The proposed method revealed the fact that the existing LLM is only performing well on low-cognitive-level tasks that mainly involve knowledge memorization. However, even the most state-of-the-art LLM can fail in high-level clinical diagnosis tasks. It also reveals the important relationship between model size and its ability in high-level complex scenarios.

**Claims And Evidence:**

The proposed medical LLM benchmark is inspired by Bloom's taxonomy, which is very intuitive and convincing. The proposed task construction protocol is reasonable and thoughtfully evaluated by human experts. The evaluation results provide intuitive but still inspiring results. Also, the QA example provided in the supplementary helps to better understand each task.

**Essential References Not Discussed:**

While the paper has a detailed discussion of existing works. It will be great if it can provide some more discussion on the other medical LLM benchmark released recently, such as [a]. But this will not harm the contribution of the paper.

[a] Zhou, Yuxuan, et al. "Reliable and diverse evaluation of LLM medical knowledge mastery." arXiv preprint arXiv:2409.14302 (2024).

**Experimental Designs Or Analyses:**

The experiment in the paper is very convincing and thorough. It covers more than 20 different general LLMs and multiple medical-specific LLMs fine-tuned from these baseline LLMs. The baselines have also covered different parameter sizes from 2 B to 70 B. The task-specific results in Tables 3 and 4 further help to understand the behavior of the medical LLMs. Overall, the experiment is convincing and reasonable. The evaluation of the medical-specific LLMs is very interesting since it helps illustrate the potential problem within these LLMs, where the post-training potentially ruined the reasoning capability of the original model, resulting a worse high-level task performance.

**Methods And Evaluation Criteria:**

The proposed novel benchmark dataset seems to be very promising and significant to the reviewer. The concerns about the LLM's performance on medical application is a long-existing problem. However, current benchmarks mainly focus on simple knowledge-based QA tasks, ignoring the complex situation in the real-world diagnosis procedure. The proposed benchmark dataset will be helpful to better evaluate the capability of existing LLMs and provide more informative results to guide the development of the field.

**Other Comments Or Suggestions:**

N/A

**Other Strengths And Weaknesses:**

Overall, the paper looks pretty solid and convincing. The proposed benchmark provides a more detailed understanding of existing LLMs. It can serve as an important step towards LLM's application in real-world diagnosis.

**Questions For Authors:**

One small thing: I am wondering if the proposed benchmark has a name or not. It will be much easier for others to refer to in the future.

**Relation To Broader Scientific Literature:**

The paper has properly discussed related literature and the status quo of current medical LLM evaluation. It can potentially serve as the standard evaluation protocol for future medical LLM evaluation, providing a more reasonable and realistic evaluation.

**Theoretical Claims:**

N/A. There is no new theoretical claims proposed in the paper.

---

> ### Author Rebuttal · Authors · 2025-04-01
>
> We sincerely appreciate your kind feedback as well as your recognition of our work. Below are our responses to each of the concerns you raised.
>
> 1. **Discussion with other LLM benchmark**: We sincerely appreciate your kind suggestions. Zhou et al. [1] proposed a medical evaluation framework to generate reliable and diverse test samples based on knowledge bases, addressing the issues of low reliability and diversity in automatic test sample generation. Meanwhile, our work aims to explore the limitations of LLMs in solving real-world medical problems by constructing a benchmark with medical tasks of varying cognitive levels and systematically evaluating existing LLMs on this benchmark, offering insights for developing LLMs suited to real-world medical applications. Once again, thank you for your kind suggestions. We will strengthen discussions with other medical LLM benchmarks in the revised paper.
>
>    [1] Zhou, Yuxuan, et al. "Reliable and diverse evaluation of LLM medical knowledge mastery." arXiv preprint arXiv:2409.14302 (2024).
>
> 2. **Name of proposed benchmark**: Thank you for your kind suggestions. Indeed, a good benchmark name could facilitate reference and discussion. We plan to name the benchmark MulCogEval (Multiple-Cognitive-Level Evaluation) and will include relevant annotations in the paper.

---

> > ### Comment · Reviewer_z2s4 · 2025-04-02
> >
> > I also apperciate the effort of the author during the rebuttal period. It is very impressive to see the additional results of human expert on the same evaluation, which can serve as an important baseline for future purpose. The new results of the Deepseek-R1 and GPT-o3 is very impressive. The performance of Deepseek-R1 is very promising and I would like to see to more discuss about the capability of the reasoning LLM and ituition of why it can work so good.
> >
> > The concerns about QA settings does not concerns me too much since the high-level idea is intuitive to me. While the key knowledge point may be the same, the changing form of the question will still increasing the difficulty of the task via different path, e.g. extended answering space, hidden information in the description, and reasoning request.
> >
> > Overall, the reviewer acknowledge the contribution and significance of this work and would like to maintain my recommendation of acceptance. A more complex and high-level medical benchmark is important to the current development of MedLLM. It is also a necessary path to approaching real-world usable MedLLMs.

---

> > > ### Author Response · Authors · 2025-04-08
> > >
> > > We would like to express our sincere appreciation for your kind and insightful feedback.
> > >
> > > 1. **About expert evaluation**: Thank you for recognizing our efforts in conducting the expert evaluation during the rebuttal phase. This evaluation not only helps validate the difficulty distinctions across cognitive levels in our benchmark, but also can serve as a meaningful baseline for future research.
> > > 2. **Discussion about reasoning LLMs**: We are grateful for your constructive suggestions. Following your kind suggestions, we further explored the possible reasons behind the strong performance of reasoning LLMs, based on the evaluation of DeepSeek-V3 and DeepSeek-R1. The evaluation results are presented below (*Relative Improvement* is calculated by dividing the absolute gain by the performance of DeepSeek-V3):
> > >
> > >    |Model| Low-Level|Mid-Level|High-Level|
> > >    |-|:-:|:-:|:-:|
> > >    |DeepSeek-V3|78.3|44.8|19.4|
> > >    |DeepSeek-R1|89.3|73.1|25.9|
> > >    |Relative Improvement|+14.0%|+63.2%|+33.5%|
> > >
> > >    We observe that although DeepSeek-R1 improves across all three levels, the gains are larger on Mid-Level tasks compared to Low-Level ones. This may be because, while both levels evaluate the same knowledge points, Mid-Level tasks are more difficult due to reduced additional information and a broader decision space. Meanwhile, reasoning LLMs can tackle these challenges by analyzing task requirements and breaking down the problem into manageable steps. For example, consider the following Answer Existence Judgment Question, where the model must determine whether a correct answer is present among the given options:
> > >
> > >    > Question: A 30-year-old woman presents to the physician because of ongoing diarrhea ... What is the possible diagnosis?
> > >    >
> > >    > A: Ulcerative colitis	B: Chronic diverticulitis	C: Acute infective colitis	D: Pseudomembranous colitis	E: Irritable bowel syndrome.
> > >
> > >    In this case, the correct diagnosis "Crohn’s disease" is not included in the options. We observe that DeepSeek-V3 incorrectly answers "yes", whereas DeepSeek-R1 arrives at the correct answer by first analyzing the patient condition, then evaluating each option based on the analysis, and finally integrating the evaluation results:
> > >
> > >    > DeepSeek-R1: Okay, so I need to figure out the diagnosis for this patient. Let's start by reviewing the case details…
> > >    >
> > >    > Then, let's list the options: … I need to check if any of these fit. UC (option A) typically presents with ... However, UC's biopsy usually ... Chronic diverticulitis (option B) often presents with ... the biopsy findings don't align with ...
> > >    >
> > >    > If the options are only the given ones, and none fit perfectly, then the correct answer is not listed. Therefore, the answer would be 'no'.
> > >
> > >    Moreover, although Low-level tasks are relatively simple and both chat & reasoning LLMs perform well at this level, we observe that reasoning LLMs can further enhance performance by generating rationale that more thoroughly link the problem to the learned knowledge.
> > >
> > >    Furthermore, compared to tasks at the other two levels, High-level task requires LLMs to actively obtain patient information through examinations and ultimately complete the diagnosis based on the gathered information, making them significantly more difficult. Notably, reasoning LLMs achieved a 33.5% improvement on this type of task. We further analyzed DeepSeek-R1’s fine-grained performance on this task, and the results are as follows:
> > >
> > >    |Model|Examination Recall|End-Point Diagnostic Acc|Full-Path Diagnostic Acc|
> > >    |-|:-:|:-:|:-:|
> > >    |DeepSeek-V3|30.0|53.6|19.4|
> > >    | **DeepSeek-R1** |**43.7**|**56.0**|**25.9**|
> > >
> > >    We observed that DeepSeek-R1 demonstrates a stronger ability (higher Exam Recall) to actively request key diagnostic information compared to DeepSeek-V3, resulting in higher full-path diagnosis accuracy on the High-Level task.
> > >
> > >    Additionally, while reasoning LLMs performed notably on our benchmark, our error analysis revealed that a significant portion of the errors stem from insufficient mastery of medical knowledge. Therefore, we suggest that current LLMs should further integrate more medical knowledge and combine it with their reasoning capabilities to more effectively address real-world medical problems.
> > >
> > > 3. **About QA settings**: Thank you for your deep understanding of our benchmark settings. Indeed, while Low- and Mid-Level tasks evaluate the same knowledge points, task format changes can further increase cognitive difficulty from different perspectives (e.g., reducing information, expanding decision space). Moreover, compared to lower-level tasks where LLMs passively receive information, High-level tasks increase cognitive difficulty by evaluating LLMs’ ability to actively plan and request key diagnostic information. Once again, we appreciate your thorough understanding and recognition of our approach and are confused that Reviewer VJ98 fails to grasp the purpose behind our design.

---

### Decision · Program_Chairs · 2025-05-01

**Decision:**

Accept (poster)

**Comment:**

This paper proposes a multi-cognitive-level evaluation framework for medical LLMs, drawing inspiration from Bloom’s Taxonomy. The core contribution is the systematic design of tasks with increasing cognitive complexity—from basic knowledge recall to scenario-based clinical reasoning—and the comprehensive evaluation of over 30 general and medical-domain LLMs. The framework is further supported by clinician-based validation, which reinforces the credibility of the cognitive-level distinctions. However, the methodology largely repurposes existing QA datasets such as MedQA, MedMCQA, and MIMIC-IV, without introducing new benchmarks or fundamentally novel evaluation paradigms. While the cognitive framing adds structure and interpretability, recent works [1,2] have already explored broader and more diverse evaluation strategies for medical LLMs. Overall, although the conceptual novelty is limited, the framework is well-executed and the empirical analysis is thorough. Given the reviewer scores and practical relevance of the proposed benchmark, I lean toward a very weak accept, but would not object if the paper is ultimately rejected.